# The variations of VOCs based on the policy change of Omicron in traffic-hub city Zhengzhou

Bowen Zhang[1, 3], Dong Zhang[2, 3], Zhe Dong[2, 3], Xinshuai Song[1, 3], Ruiqin Zhang[1, 3], Xiao Li[1, 3,*]

[1]School of Ecology and Environment, Zhengzhou University, Zhengzhou 450001, China

[2]College of Chemistry, Zhengzhou University, Zhengzhou 450001, China

[3]Institute of Environmental Sciences, Zhengzhou University, Zhengzhou 450001, China

Correspondence to: Xiao Li, E-mail address: lixiao9060@zzu.edu.cn

**Abstract:** Online volatile organic compounds (VOCs) were monitored before and after the Omicron policy change at an urban site in polluted Zhengzhou from December 1, 2022, to January 31, 2023. The characteristics and sources of VOCs were investigated. The daily mean concentrations of $PM_{2.5}$ and total VOCs (TVOCs) ranged from 53.5 to 239.4 µg/m³ and 15.6 to 57.1 ppbv, respectively, with mean values of 111.5 ± 45.1 µg/m³ and 36.1 ± 21.0 ppbv, respectively, throughout the period. Two severe pollution events (designated as Case 1 and Case 2) were identified in accordance with the National Ambient Air Quality Standards (NAAQS) (China's National Ambient Air Quality Standards (NAAQS) from 2012). Case 1 (December 5 to December 10, $PM_{2.5}$ daily mean = 142.5 µg/m³) and Case 2 (January 1 to January 8, $PM_{2.5}$ daily mean = 181.5 µg/m³) occurred during the infection period (when the policy of "full nucleic acid screening measures" was in effect) and the recovery period (after the policy was cancelled), respectively. The $PM_{2.5}$ and TVOCs values for Case 2 are, respectively, 1.3 and 1.8 times higher than those for Case 1. The precise influence of disparate meteorological circumstances on the two pollution incidents is not addressed in this study. The results of the positive matrix factor modeling demonstrated that the primary source of volatile organic compounds (VOCs) during the observation period was industrial emissions, which constituted 32% of the total VOCs, followed by vehicle emissions (27%) and combustion (21%). In Case 1, industrial emissions constituted the primary source of VOCs, accounting for 32% of the total VOCs. In contrast, in Case 2, the contribution of vehicular emission sources increased to 33% and became the primary source of VOCs. The secondary organic aerosol formation potential for Case 1 and Case 2 were found to be 37.6 µg/m³ and 65.6 µg/m³, respectively. In Case 1, the

largest contribution of SOAP from industrial sources accounted for the majority (63%,
23.8 μg/m³), followed by vehicular sources (18%). After the end of the epidemic and
the resumption of productive activities in the society, the difference in the proportion
of SOA generated from various sources decreased. Most of the SOAP came from
solvent use and fuel evaporation sources, accounting for 32% (20.9 μg/m³) and 26%
(16.8 μg/m³), respectively. On days with minimal pollution, industrial sources and
solvent use remain the main contributors to SOA formation. Therefore, regulation of
emissions from industry, solvent-using industries and motor vehicles need to be
prioritized to control the PM$_{2.5}$ pollution problem.

**Keywords: Volatile organic compounds; Pollution episode; Source apportionment; Positive**
**Matrix Factorization model; Secondary organic aerosol formation potential;**

## 1. Introduction

Volatile organic compounds (VOCs) in the atmosphere have high reactivity and can react with nitrogen oxides ($NO_x$) to form a series of secondary pollutants such as ozone ($O_3$) and secondary organic aerosol (SOA), resulting in regional air pollution (Li et al., 2019; Hui et al., 2020). The problem of $O_3$ pollution has been plaguing major urban agglomerations in China (Zheng et al., 2010; Li et al., 2014; Wang et al., 2017). SOA is an important component of fine particulate matter ($PM_{2.5}$) and contributes significantly to haze pollution (Liu et al., 2019). $PM_{2.5}$ remains the most significant air pollutant in many Chinese cities for years (Shao et al., 2016; Wu et al., 2016). In addition, VOCs, represented by the benzene homologues, can cause damage to kidneys, liver, and nervous system of humans when they enter the body (Zhang et al., 2018).

Studies have shown that the most common VOC components in China are alkanes, olefins, aromatic hydrocarbons, oxygenated VOCs (OVOCs), and halogenated hydrocarbons, among which alkanes are the most abundant species (Liu et al., 2020; Zhang et al., 2021a). VOCs in the atmosphere have a wide range of sources, and VOCs in different regions are affected by multiple factors such as local geography, climate, and human activities (Mu et al., 2023; Zou et al., 2023). The above reasons lead to significant regional and seasonal differences in the characteristics of VOCs (Song et al., 2021). For example, the annual average concentration of VOCs in the coastal background area of the Pearl River Delta is 9.3 ppbv. The seasonal variation trend of VOCs is high in autumn and winter and low in summer (Yun et al., 2021). In contrast, the average VOC concentration in autumn and winter in Beijing was $22.6 \pm 12.6$ ppbv, and the VOC concentration in the winter heating period was twice that in the autumn non-heating period (Niu et al., 2022).

Moreover, the sources of VOC components in different regions are also related to the local industrial structure and living habits. In rural areas of North China Plain in winter, it is found that the SOA formation potential (SOAP) of VOCs under low $NO_x$ conditions is significantly higher than that under high $NO_x$ conditions, and the increase of aromatic hydrocarbon emissions caused by coal combustion is the main reason for the higher SOAP in winter (Zhang et al., 2020). Li et al. (2022) found that the average increased concentration of acetylene was 4.8 times from autumn to winter in the Guanzhong Plain, indicating that fuel combustion during the heating period in winter has a significant impact on the composition of VOCs. In contrast, continuous

observations conducted by Zhou et al. (2022) in the suburbs of Dongguan in summer found that industrial solvent usage, liquefied petroleum gas (LPG) and oil and gas volatilization were the main sources of VOCs. The results highlighted a wide variation of characteristics, sources and chemical reactions of VOCs in the atmosphere thus it is necessary to investigate VOCs in different cities when formulating control measures.

Zhengzhou, as the capital of Henan Province, is an important transportation hub and economic center in the Central Plains region. Zhengzhou is currently facing significant air pollution problems, with the Air Quality Index at the bottom of the national ranking of 168 cities for many years. In January 2023, for example, the number of polluted days with $PM_{2.5}$ as the primary pollutant was 17, and the daily average value of $PM_{2.5}$ reached a maximum of 298 $\mu g/m^3$ (https://www.aqistudy.cn/historydata/daydata.php?city=%E9%83%91%E5%B7%9E &month=202301, Accessed Jan 2024), which is almost 300% higher than the Chinese daily average standard (grade II, 75 $\mu g/m^3$). The studies of VOCs were carried out in Zhengzhou in recent years, which focused on the characteristics and sources of VOCs during pollution episodes (Lai et al., 2024) or before the coronavirus epidemic outbreak (Li et al., 2020; Zhang et al., 2021b). While some atmospheric VOCs studies involving the impact of Covid-19 lockdown have been performed in India (Singh et al., 2023a), in China (e.g., Pei et al., 2022; Jensen et al., 2023; Zuo et al., 2024), or with respect to toluene, benzene, m/p-xylene and ethylbenzene only (e.g., Sahu et al., 2022; Singh et al., 2023b), a gap persisted in the investigation of VOCs due to the impact of abolishment of China's zero-policy. Furthermore, some studies have discussed the impact of changes in human production activities on air pollution during and after the outbreak of the coronavirus disease (e.g., Ma et al., 2022; Jiang et al., 2023; Song et al., 2023), but as mentioned earlier, only a few studies with in-depth exploration of the changes in VOCs and none dealing with ending the zero-Covid policy during Omicron variant infection period.

In this study, we conducted continuous online observations of VOCs during the polluted winter season at an urban site in Zhengzhou. The study covered the period following the removal of lockdown measures. We focused on pollution events when the daily average $PM_{2.5}$ concentration exceeded 75 $\mu g/m^3$ (China's Class II standard) for more than three consecutive days. Days with $PM_{2.5}$ concentrations below 35 $\mu g/m^3$ (China's Class I standard) were classified as clean days. During this period, China lifted zero-COVID strategies, announcing the '10 measures' for optimizing COVID-19 rules

on December 7, 2022 (http://www.news.cn/politics/2022-12/07/c_1129189285.htm,
Accessed Jan 2024). Zhengzhou's epidemic prevention and control measures changed
with the issuance of Circular No. 163 on December 4, 2022, which allowed the
reopening of closed public places. As a result, movement within Zhengzhou increased
and social production resumed. Our research specifically examines the period
dominated by the COVID-19 Omicron variant. where they demonstrate notable
differences from the early virus strains (i.e., original SARS-CoV-2 virus and Delta) in
terms of geographical transmission, the scale of the infected population, and symptom
manifestation (Petersen et al., 2022; Merino et al., 2023).
After the quarantine policy was lifted, people basically rested at home due to
infection or fear of infection with Omicron. The resumption of normal production and
life depends on herd immunization. This outbreak event is the longest in duration and
the largest in number of infections since the 2020 outbreak of the novel coronavirus in
Zhengzhou. It would be beneficial to investigate the impact of this event on emissions
related to transportation and industrial production. This change is worth exploring in
terms of its impact on transportation and industrial production emissions. Therefore,
the characteristics and variations of VOCs during different periods were investigated to
assess their impact on the formation of SOA and to provide data support for future
pollution control policies in Zhengzhou.

## 2. Materials and methods

### 2.1 Sample collection and Chemical analysis

The online VOCs observation station is located on the roof of the Zhengzhou Environmental Protection Monitoring Center, which is in the urban area. The sampling site is close to main roads on three sides (150 m away from Funiu Road on the east side, 200 m away from Qinling Road on the west side, and connected to Zhongyuan Road on the south side), and surrounded by residential areas and commercial areas without other large nearby stationary sources. The sampling period for this study was from December 1, 2022, to January 31, 2023, and serious $PM_{2.5}$ pollution in Zhengzhou was of frequent occurrence during December and January. (https://www.aqistudy.cn/historydata/monthdata.php?city=%e9%83%91%e5%b7%9e #:~:text=%E7%94%9F%E5%91%BD%E6%9D%A5%E6%BA%90%E8%87%AA% E7%84%B6%EF%BC%8C%E5%81%A5). Apart from a brief occurrence of rain and snow on December 25, the sampling days were either sunny or cloudy. The wind speed (WS), temperature (Temp) and relative humidity (RH) during this period were $1.3 \pm 0.9$ m/s, $5.3 \pm 3.2$ ℃ and $38.9 \pm 19.0\%$), respectively, similar to the values observed in previous years in Zhengzhou. It is interesting to point out that the sampling period in the present study covered the entire infection period of Omicron in Zhengzhou, including the phase of surge in infected population (Infection period, from 2022.12.01 to 2022.12.31) and restoration of production and livelihood phase (Recovery period, from 2023.1.1 to 2023.1.31 in 2023) (Fig. S1, Chinese Center for Disease Control and Prevention, 2023).

The VOCs were measured hourly using a GC-FID/MS (TH-PKU 300 b, Wuhan Tianhong Instruments Co., China). The instrument TH-PKU300b includes electronic refrigeration ultra-low temperature pre-concentration sampling system, analysis system and system control software. The ambient VOCs in the first 5 minutes of each hour were collected by the sampling system and then entered the concentration system. Under low temperature conditions, the VOCs samples collected were frozen in the capillary capture column, and then quickly heated and resolved, so that the compounds entered the analysis system. After separation by chromatographic column, the compounds were monitored by FID and MS detectors. During the detection process, the atmospheric samples collected undergo analysis through two distinct pathways. C2-C5 hydrocarbons are analyzed using FID, while C5-C12 hydrocarbons, halocarbons,

and OVOCs are analyzed with a MS detector. After excluding species with missing data exceeding 10%, the detected volatile organic compounds include 29 alkanes, 11 alkenes, 17 aromatics, 35 halocarbons, 12 OVOCs, 1 alkyne (acetylene), and 1 sulfide ($CS_2$) with a total of 106 compounds. A detailed description of the instrumentation can be found in our previous study (Zhang et al., 2021b; Shi et al., 2022; Zhang et al., 2024).

The instrument was calibrated per week to ensure the accuracy of VOCs by injecting standard gases with a five-point calibration curve. The detection limit of C2-C5 hydrocarbons ranges from 0.007 to 0.099 ppbv, other hydrocarbons are 0.004–0.045 ppbv, halogenated hydrocarbons 0.009-0.099 ppbv, OVOCs and other compounds of 0.006–0.095 ppbv. Thirty-two of the monitored VOCs had over 90% observed data greater than the detection limit, and 34 had more than 50% observed data greater than the detection limit.

Simultaneous observations at the same site were also carried out for particulate matter ($PM_{2.5}$, $PM_{10}$), other trace gases (carbon monoxide (CO), $O_3$, nitric oxide (NO), nitrogen dioxide ($NO_2$)), and meteorological data (Temperature, RH, WS, and wind direction (WD)) based on 1 h resolution.

## 2.2 Positive Matrix Factorization (PMF) model

EPA PMF5.0 model was used for the quantitative source analysis of VOCs (Norris et al., 2014). The principles and methods have been described in detail in previous studies (Mozaffar et al., 2020; Zhang et al., 2021b). The decomposition of the PMF mass balance equations is simplified as follows (Norris et al., 2014):

$$x_{ij} = \sum_{k=1}^{p} g_{ik} \, f_{kj} + e_{ij} \qquad (1)$$

where $x_{ij}$ is the mass concentration of species $j$ measured in sample $i$; $g_{ik}$ is the contribution of factor $k$ to the sample $i$ ; $f_{kj}$ represents the content of the $j$th species in factor $k$; $e_{ij}$ is the residual of species $j$ in sample $i$; $p$ represents the number of factors. The fitting objective of the PMF model is to minimize the function $Q$ to obtain the factor contributions and contours. The formula for $Q$ is given in Eq. (2):

$$Q = \sum_{i=1}^{n} \sum_{j=1}^{m} \left[ \frac{x_{ij} - \sum_{k=1}^{p} g_{ik} f_{kj}}{u_{ij}} \right]^2 \qquad (2)$$


where $n$ and $m$ denote the number of samples and VOC species, respectively.

Concentrations and uncertainty data are required for the PMF model. In this study,

the median concentration of a given species is used to replace missing values with an
uncertainty of four times of the median values; data less than the Method Detection
Limit (MDL) were replaced with half the MDL, with an uncertainty of 5/6 of the MDL;
and the uncertainty for values greater than the MDL was calculated using Eq. (3). In
Eq. (3), $EF$ is error fraction, expressed as the precision of VOCs species, and the setting
range can be adjusted from 5 to 20% according to the concentration difference (Buzcu
et al., 2006; Song et al., 2007); and $c_{ij}$ is the concentration of species $j$ in sample $i$:
$$U_{ij} = \sqrt{\left( EF \times c_{ij} \right)^2 + (0.5 \times MDL)^2} \qquad (3)$$

when the concentration of VOCs in the species is less than the value of the

detection limit $U_{ij}$ is calculated using Eq. (4):
$$U_{ij} = \left( \frac{5}{6} \right) MDL \qquad (4)$$

VOC species and concentration input into PMF were carefully selected to ensure

the accuracy of the PMF results. Species were excluded when over 25% of the samples
were missing or concentrations values were below the MDL (Gao et al., 2018); VOCs
with a short lifetime in the atmosphere were also excluded unless they are source-
relative species (Zhang et al., 2014; Shao et al., 2016). After that, retained VOC species
were categorized according to the signal-to-noise ratio (S/N) with S/N < 0.2 species
categorized as bad, 0.2 < S/N < 2 species categorized as weak; and S/N > 2 species
categorized as strong (Shao et al., 2016).

We used displacement of factor elements (DISP) to assess PMF modelling

uncertainty (for a description, see Paatero et al. (2014)). Q was less than 1% and no
swaps occurred for the small est dQ$^{max}$ in DISP. Fpeak values from -2 to 2 were tested
to explore the rotational stability of the solutions. $Q_{true}/Q_{exp}$ is lowest when Fpeak = 0,
so we chose the PMF results for that case (Fig. S2a). After examining 3-8 factors, 20
base runs with 5 factors eventually selected to represent final result. We provide an
explanation of factor selection in the supplementary materials. Fig. S2(b) includes
$Q_{true}/Q_{exp}$, $Q_{robust}/Q_{exp}$ for factors 3-8. The slopes of these two ratios in changed at five
factors, and we found that five factors were more realistic after repeated comparisons
of the results at four, five and six factors.
**2.3 SOA generation potential**
The contributions of VOC species to SOAP were calculated based on the toluene
weighted mass contributions (TMC) method (Derwent et al., 2010). The methodology
for calculating SOAP is as follows:

$$SOAPF_i = \frac{VOCs\ component\ i\ to\ SOA\ mass\ concentration\ increments}{Toluene\ to\ SOA\ mass\ concentration\ increment} \times 100 \quad (5)$$

$SOAPF_i$ for each VOC is taken from the literature (Derwent et al., 2010). The
SOAP was estimated by multiplying the $SOAPF_i$ value by the concentration of
individual VOC species. The SOAP calculations through each VOC are as follows:

$$SOAP = \sum E_i \times SOAPF_i \quad (6)$$
$E_i$ is the concentration of species $i$.

**3. Results and discussion**
**3.1 Overview of variation in pollutants and meteorological**
**parameters**
Figure 1 shows the time series of meteorological parameters, TVOCs, $O_3$, $NO_x$,
$SO_2$, CO and $PM_{2.5}$ during the observed periods. Low WS and Temperature were found
with an average value of $1.3 \pm 0.6$ m/s and $5.0 \pm 2.5$ °C, respectively, during the entire
period, comparable with observations at the same site in 2021 (Lai et al., 2024). A total
of 62 days of valid data was acquired with the daily average concentration of $PM_{2.5}$
ranging from 53 to 239 $\mu g/m^3$, with the average value of $111 \pm 45$ $\mu g/m^3$. The
concentration of TVOCs ranged from 15.6 to 57.1 ppbv with an average of $36.1 \pm 21.0$
ppbv, higher than the same period in last year ($27.9 \pm 12.7$ ppbv, Lai et al., 2024).
During the observation period, the average values of T, WS and RH were $5.0 \pm 2.5$ °C,
$1.3 \pm 0.6$ m/s and $38.9 \pm 16.7\%$, respectively.
Previous studies have shown that meteorological factors such as low WS, high RH,
and low precipitation are responsible for the increase in $PM_{2.5}$ pollution in Zhengzhou
in winter (Duan et al., 2019). Our analysis of the correlation between different
pollutants and meteorological conditions during the pollution period showed that
$PM_{2.5}$, TVOCs and $NO_x$ were positively correlated with relative humidity (Fig. S3),
which is consistent with the results of some previous studies (Wang et al., 2019). The
comparisons of average concentrations of different periods between different periods
are presented in Tables 1 and 2. In this study, the WS on clean days ($1.4 \pm 0.8$ m/s)
was higher than in Case 1 ($1.2 \pm 0.9$ m/s) and Case 2 ($0.9 \pm 0.7$ m/s), while the RH
was lower by 26.2% and 12.5% compared to Case 1 and Case 2, respectively. These
findings indicate that high RH and low WS significantly influence the occurrence of
pollution during the observation period. WS, Temp and RH conditions during
infection and recovery periods were generally similar, and meteorology may also have
played a role in the differences between pollution events, but its specific influence was
not determined here. The average concentration of $PM_{2.5}$ during the recovery period
was 1.6 times the value during the infection period. Furthermore, the concentrations
of other pollutants including $SO_2$, $NO_2$, CO, and $O_3$ all showed a similar trend between
infection and recovery periods. The TVOC concentration during the recovery period
was 1.2 times the value during the infection period, showing an obvious increase trend
after resuming production. Decreased trends of air pollutants were found in other
studies before and after the outbreak of the novel coronavirus (COVID-19) in early
2020 (Qi et al., 2021; Wang et al., 2021).

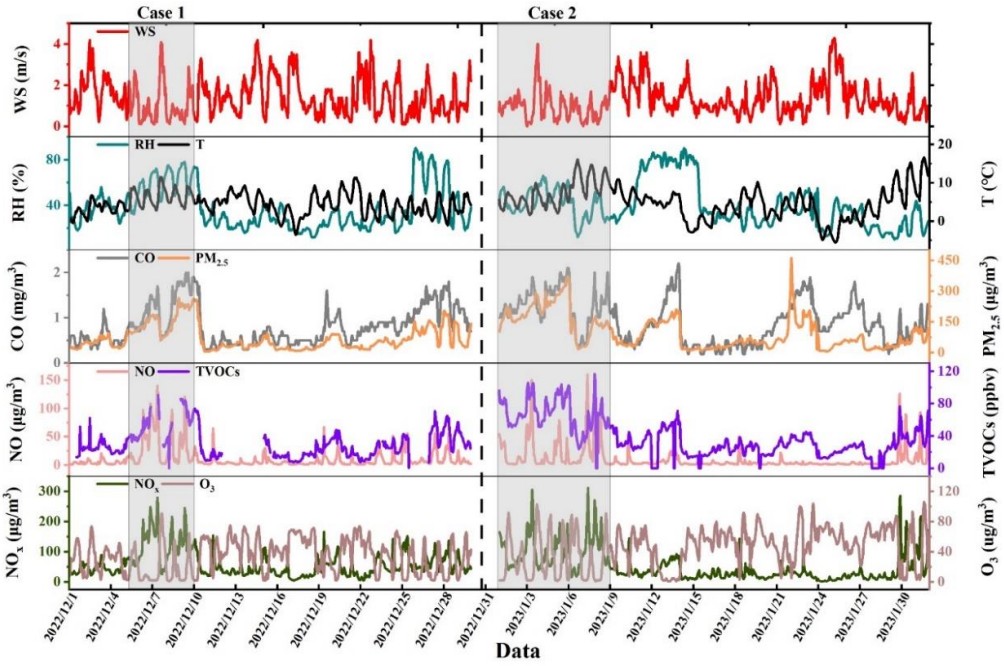


Fig. 1. Time series of WS, T, RH, CO, $PM_{2.5}$, NO, TVOCs, $NO_x$ and $O_3$ during the observation

period.

The shadow section in Fig. 1 represents two haze pollution events during the

monitoring period. A pollution event is determined when the daily average
concentration of $PM_{2.5}$ exceeds 75 $\mu g/m^3$ (China's II-level standard) for at least three
consecutive days. Case 1 (December 5 to December 10 with daily average $PM_{2.5}$ =
142.5 $\mu g/m^3$) and Case 2 (January 1 to January 8 with daily average $PM_{2.5}$ = 181.5
$\mu g/m^3$) were selected as they represent the pollution events in infection and recovery
periods, respectively, due to their long duration and high pollution levels. Any day with
a $PM_{2.5}$ concentration lower than 35 $\mu g/m^3$ (China's I-level standard) is considered as
Clean day.

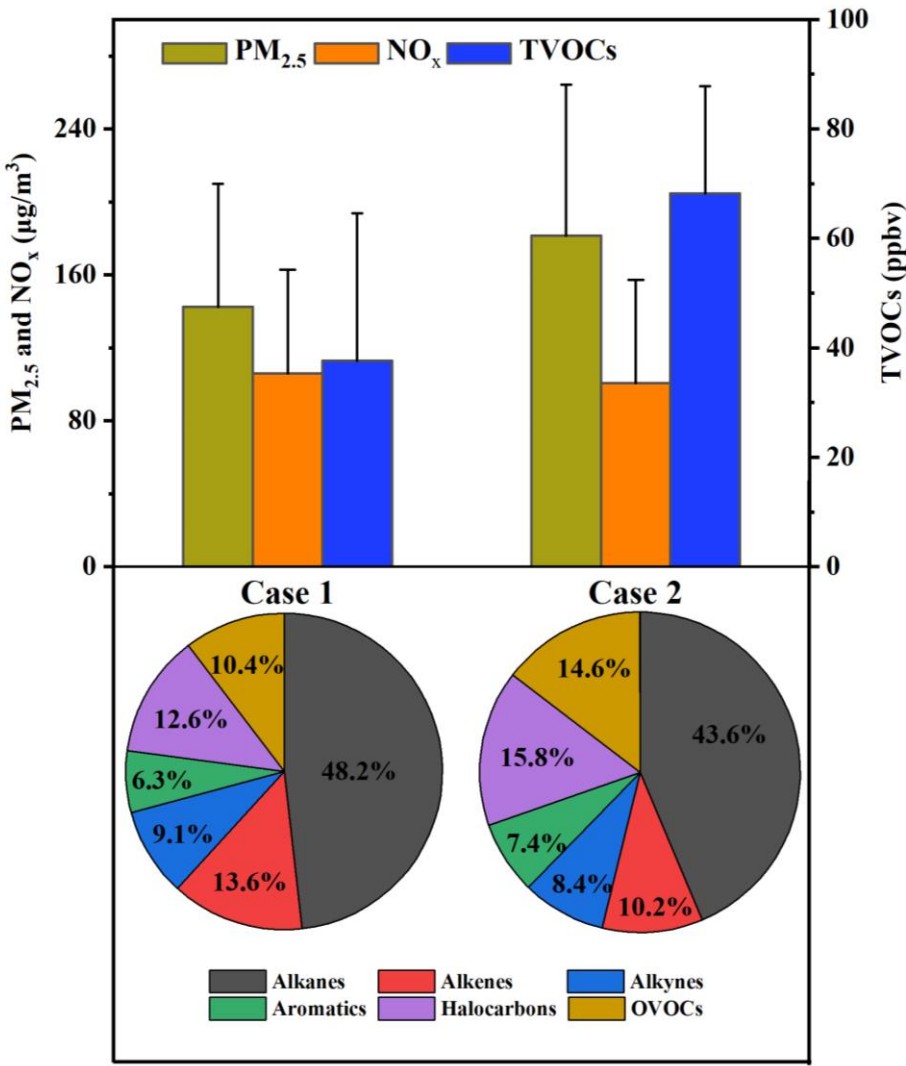

Fig. 2. The concentration of PM$_{2.5}$, NO$_x$, TVOCs and the composition ratio of VOCs in Case 1 and Case 2.

As for the two representative pollution processes (Case 1 during the infection period and Case 2 during the recovery period), the concentration of TVOCs in Case 1 and Case 2 were 48.4 ± 20.4 and 67.6 ± 19.6 ppbv (Fig. 2), respectively, increased by 63% and 188% compared with values during clean days. The average concentrations of PM$_{2.5}$ and TVOCs during Case 2 were 1.3 and 1.8 times the values in Case 1. The highest volume contributions of alkanes were found both in Case 1 (48%) and Case 2 (44%), consistent with the results in the Yangtze River Delta region (36-43%, Liu et al., 2023). While alkenes exhibited higher volume percentages of 13% in Case 1, followed by halogenated hydrocarbon (12%) and OVOCs (10%). Higher volume percentages of alkanes and alkenes in Case 1 were similar to the results in the gasoline evaporation site in winter (Niu et al., 2022). Equivalent volume contribution of halogenated hydrocarbon and OVOCs (15%) were found in Case 2, followed by alkenes (10%).

Although aromatic hydrocarbons have the lowest volumetric contribution (6% in Case 1 and 7% in Case 2), they show the largest increase from clean days to pollution.

Table 1 The average concentrations of meteorological parameters and pollutants during different processes.

| Category | Entire process (2022.12.1-2023.1.31) | Infection period (2022.12.1-2022.12.31) | Recovery period (2023.1.1-2023.1.31) | Case 1 (2022.12.5-2022.12.10) | Case 2 (2023.1.1-2023.1.8) | Clean Days |
|---|---|---|---|---|---|---|
| | N = 62 days | N = 31 days | N = 31 days | N = 6 days | N = 8 days | N = 8 days |
| WS (m/s) | $1.3 \pm 0.6$ | $1.4 \pm 0.6$ | $1.3 \pm 0.6$ | $1.2 \pm 0.9$ | $0.9 \pm 0.7$ | $1.4 \pm 0.8$ |
| T (°C) | $5.0 \pm 2.5$ | $4.7 \pm 1.7$ | $5.4 \pm 3.1$ | $6.1 \pm 2.2$ | $7.4 \pm 3.5$ | $4.1 \pm 3.0$ |
| RH (%) | $38.9 \pm 16.7$ | $37.6 \pm 15.5$ | $40.2 \pm 18.2$ | $55.7 \pm 14.7$ | $42.0 \pm 12.1$ | $29.5 \pm 18.1$ |
| TVOCs (ppbv) | $36.1 \pm 21.0$ | $31.9 \pm 18.1$ | $39.8 \pm 22.4$ | $37.6 \pm 27.0$ | $68.2 \pm 19.6$ | $22.7 \pm 11.1$ |
| $SO_2$ ($\mu g/m^3$) | $11.4 \pm 2.7$ | $10.2 \pm 2.8$ | $12.7 \pm 2.3$ | $11.0 \pm 3.7$ | $16.2 \pm 6.1$ | $6.5 \pm 2.5$ |
| $NO_2$ ($\mu g/m^3$) | $47.2 \pm 10.0$ | $46.8 \pm 8.6$ | $47.8 \pm 11.7$ | $62.7 \pm 20.5$ | $65.0 \pm 21.3$ | $20.8 \pm 15.9$ |
| CO ($mg/m^3$) | $0.9 \pm 0.2$ | $0.8 \pm 0.2$ | $1.1 \pm 0.2$ | $1.2 \pm 0.5$ | $1.3 \pm 0.4$ | $0.5 \pm 0.2$ |
| $O_3$ ($\mu g/m^3$) | $34.9 \pm 6.0$ | $31.1 \pm 4.5$ | $39.0 \pm 4.6$ | $21.8 \pm 23.7$ | $32.5 \pm 29.6$ | $52.6 \pm 18.4$ |
| $PM_{2.5}$ ($\mu g/m^3$) | $111.5 \pm 45.1$ | $86.6 \pm 34.6$ | $138.3 \pm 39.6$ | $142.5 \pm 67.4$ | $181.5 \pm 82.7$ | $23.8 \pm 16.8$ |

Table 2 Concentration of VOC species during different processes (ppbv).

| Category | Entire process | Infection period | Recovery period | Case 1 | Case 2 | Clean days |
|---|---|---|---|---|---|---|
| TVOCs | $36.1 \pm 21.0$ | $31.9 \pm 18.1$ | $39.8 \pm 22.4$ | $48.4 \pm 20.4$ | $67.6 \pm 19.6$ | $17.5 \pm 9.5$ |
| alkanes | $16.8 \pm 9.2$ | $15.0 \pm 8.4$ | $18.4 \pm 9.5$ | $23.1 \pm 10.0$ | $29.5 \pm 8.4$ | $9.2 \pm 5.6$ |
| alkenes | $4.1 \pm 2.7$ | $3.8 \pm 2.6$ | $4.4 \pm 2.7$ | $6.5 \pm 2.9$ | $7.0 \pm 2.6$ | $1.7 \pm 1.3$ |
| alkynes | $3.1 \pm 2.0$ | $2.7 \pm 1.7$ | $3.4 \pm 2.1$ | $4.3 \pm 2.0$ | $5.8 \pm 1.9$ | $1.3 \pm 0.8$ |
| aromatics | $2.1 \pm 2.0$ | $1.8 \pm 1.5$ | $2.3 \pm 2.2$ | $3.0 \pm 1.8$ | $4.9 \pm 2.8$ | $0.7 \pm 0.5$ |
| halogenated hydrocarbon | $5.4 \pm 3.3$ | $4.4 \pm 2.3$ | $6.2 \pm 3.8$ | $6.0 \pm 1.9$ | $10.7 \pm 3.6$ | $2.7 \pm 1.4$ |
| OVOCs | $4.6 \pm 3.2$ | $3.5 \pm 2.7$ | $5.1 \pm 3.5$ | $5.0 \pm 2.4$ | $9.7 \pm 2.8$ | $1.9 \pm 1.1$ |

## 3.2 Source Analysis of VOCs

Specific VOC ratios can be used for initial source identification of VOCs and determination of photochemical ages of air masses (Monod et al., 2001; An et al., 2014; Li et al., 2019). In this study, the ratios of toluene/benzene (T/B), isopentane/n-pentane, isobutane/n-butane, and m/p-xylene/ethylbenzene (X/E) were selected to initially identify the potential sources of VOCs (Fig. 3). Concentrations of selected pollutants and ratios used are shown in Table S1.

The toluene-to-benzene ratio (T/B ratio) was widely used to assess the relative
importance of different sources. Specifically, T/B ratio with a value of 1.3-3.0 was
observed in vehicle emissions for vehicles with different fuel types (Schauer et al., 2002;
Wang et al., 2015). The reported T/B ratio for combustion processes was between 0.13
and 0.7 (Li et al., 2011; Wang et al., 2014). The mean value of T/B ratio for the entire
period was 1.0, with the majority of the data (99%) falling between 0.1 and 3.0 and
concentrated within the 0.7-1.3 range (49%). This suggests that both traffic emissions
and combustion may be significant sources of VOCs. It should be noted that this
analytical approach is not without limitations. The ratios observed here do not exclude
linear combinations from other sources. Consequently, an in-depth examination of the
sources of VOCs was conducted using the PMF model in the next section.
The isopentane/n-pentane concentration ratios of 0.6-0.8 represent mainly coal
combustion emissions, ratios of 0.8-0.9 represent LPG emissions, 2.2-3.8 represent
vehicle exhaust emissions, and 1.8-4.6 represent fuel evaporation (Conner et al., 1995;
Liu et al., 2008; Li et al., 2019). The sources of isopentane and n-pentane in this study
were intricate and multifaceted. The mean isopentane/n-pentane ratio was 1.4, with the
majority of data points (99%) falling within the range of 0.1-4.6, with a notable
concentration in the 0.8 to 1.8 interval. This indicates that pentane is susceptible to a
combination of LPG emissions and fuel evaporation. However, the proportion of
pentane may also be affected by a combination of coal combustion emissions and
vehicle exhaust.
Isobutane/n-butane concentration ratios of 0.2-0.3 represent vehicle emissions,
0.4-0.6 represent LPG usage, and 0.6-1.0 represent natural gas emissions (Russo et al.,
2010; Zheng et al., 2018). The mean isobutane/n-butane ratio in this study was 0.5, with
the majority of data points (99%) falling within the 0.4-0.6 range, indicating that VOCs
at the observation sites were significantly influenced by the use of LPG. (Shao et al.,
2016; Zeng et al., 2023). This result can also be caused by a combination of vehicle
exhaust and natural gas emissions.
The ratio of X/E can be used to infer the photochemical age of the air mass. X/E
ratios around 2.5-2.9 are typical of urban areas, indicating that VOCs are mainly from
the urban area (fresh air mass) (Kumar et al., 2018). When this ratio is significantly
lower than 3.0, it indicates that VOCs are mainly transported from distant sources
(aging air masses) (Kumar et al., 2018). The average X/E value in this study was 2.0
(Fig. 3(d)), indicating low photochemical activity and aging of the air mass at the
observation site. Potential source analyses also indicate that air masses are affected by
long-range transport (Fig. S4).

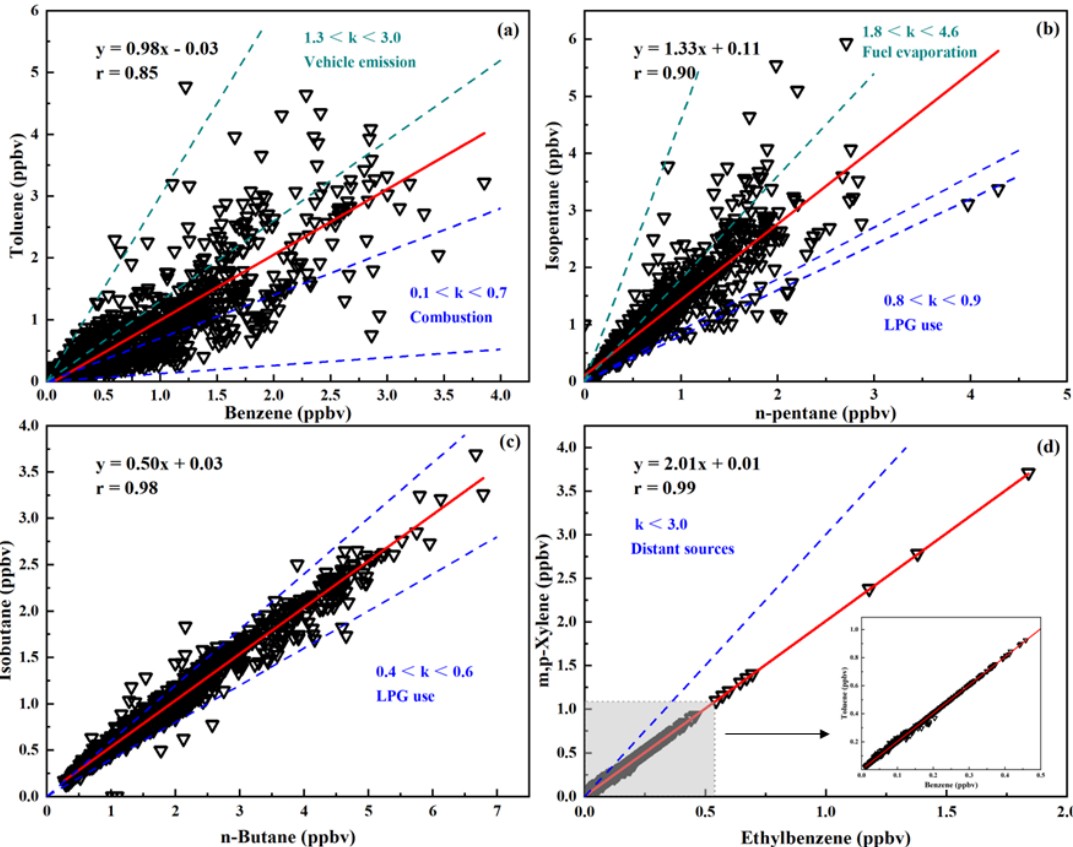


Fig. 3. Correlation analysis between specific VOC species.
Figure 4 shows the chemical profiles of individual VOCs resolved by the PMF
model during the entire observation period. These five factors eventually selected as
potential sources for the observed VOCs are: (1) Fuel evaporation; (2) Solvent usage;
(3) Vehicular emission; (4) Industrial source; and (5) Combustion. These 5 factors have
been commonly reported before, e.g., in Shijiazhuang, northern China (Guan et al, 2023)
and in Beijing (Cui et al., 2022).
Alkanes of C4-C6 substances were predominant in factor 1, including 2-
methylpentane, 3-methylpentane, isobutane, n-butane, isopentane and n-pentane from
oil and gas (Xiong et al., 2020). Fig. S5 shows that emissions from this source peak at
midday, when fuel volatilization is high, The CPF plot shows that south-east is the
dominant direction at wind speeds of less than 2 m/s (Fig. 5a). Therefore, factor 1 was
identified as the source of oil and gas volatilization.
The contribution of benzene, toluene, methylene chloride, 1,2-dichloroethane and
ethyl acetate was high in factor 2. It has been shown that benzene, toluene, ethylbenzene,
and xylene is an important component in the use of solvents (Li et al., 2015); methylene
chloride is often used as a chemical solvent, while esters are mostly used as industrial
solvents or adhesives (Li et al., 2015). Factor 2 is determined to be solvent usage source.
The CPF plot shows that due east is the main emission direction at wind speeds less
than 2 m/s and southeast is the main source at wind speeds greater than 2 m/s (Fig. 5b).
Factor 3 contains predominantly C3-C8 alkanes, olefins and alkynes, and
relatively high concentrations of benzene. These substances are usually emitted by
industrial processes (Shao et al., 2016), so Factor 4 is defined as an industrial source.
The CPF plots indicate that a local source at low wind speeds is the dominant sources
(Fig. 5c).
Factor 4 is characterized by relatively high levels of C2-C6 low-carbon alkanes
(ethane, propane, isopentane, n-pentane, isobutane and n-butane), olefins (ethylene and
propylene), and benzene and toluene, which are important automotive exhaust tracers
(Song et al., 2021; Zhang et al., 2021b). Ethylene and propylene are important
components derived from vehicle-related activities. Previous studies of VOCs in
Zhengzhou have shown a high percentage of VOCs emitted from gasoline vehicles,
with the main source of alkanes being on-road mobile sources (Bai et al., 2020). The
daily variation of this source in Fig. S5 shows a bimodal trend, with peaks occurring in
the morning and evening peaks of traffic, consistent with motor vehicle emissions. Fig.
5d shows that this source is mainly from the west where wind speeds are below 2 m/s,
and in this direction, there are a number of urban arterial roads with high traffic volumes.
Therefore, factor 4 was defined as vehicular emission source.
The highest contribution to factor 5 is chloromethane (62%). Benzene (46%) and
acetylene (41%) also contribute highly to factor 5. Chloromethane is the key tracer for
biomass combustion and acetylene is the key tracer for coal combustion (Xiong et al.,
2020). Therefore, factor 5 is defined as a combustion source. The CPF plot shows that
at wind speeds below 2 m/s, the north-east direction is the dominant source direction
(Fig. 5e).

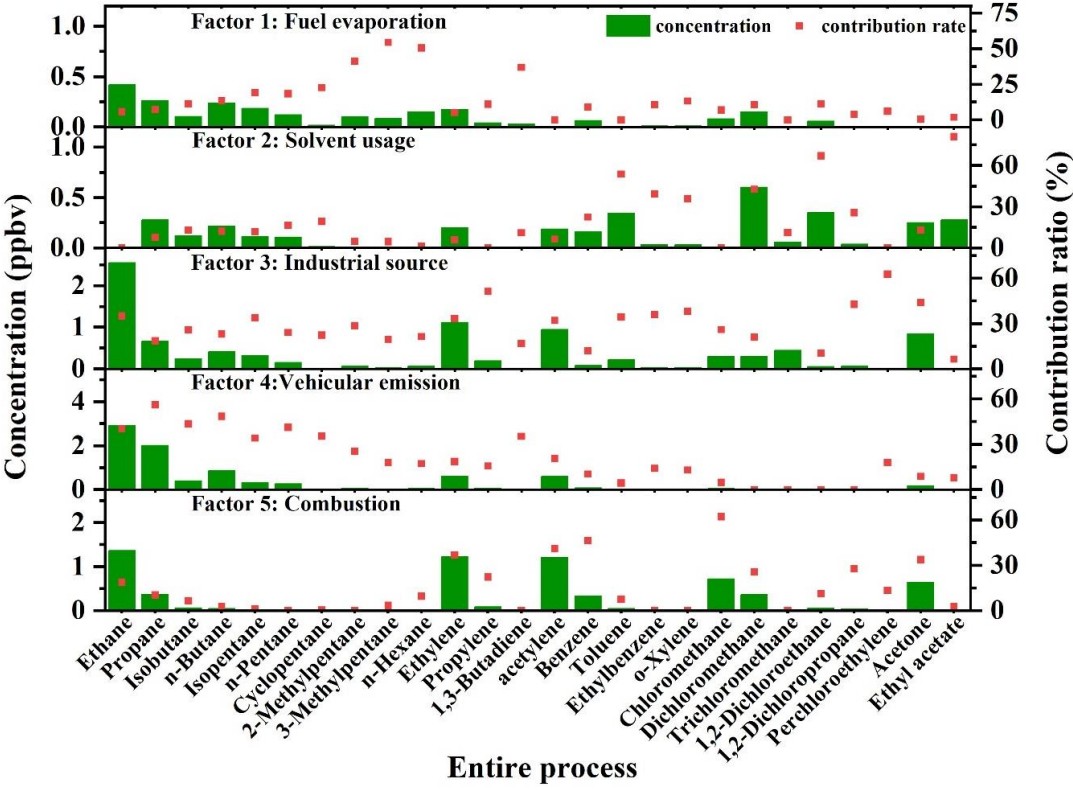


Fig. 4. Concentration of VOC species in each factor and contribution to each source.

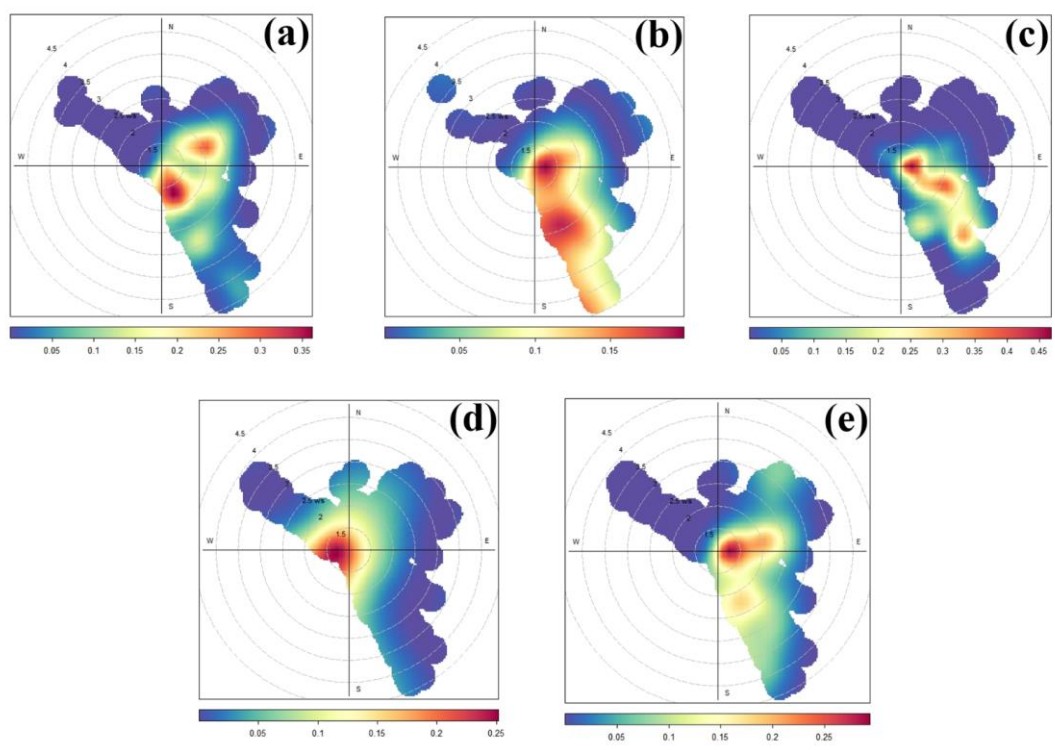


Fig. 5. CPF plots of five VOCs sources obtained using the PMF model.

Note: a: Fuel evaporation; b: Solvent usage; c: Industrial source; d: Vehicular emission; e: Combustion.

Fig. S6 compares the differences in PMF source profiles between the Omicron infection period and the recovery period, as well as between the pollution day and the clean day. We present the concentrations of the five main VOCs in all five factors in Table S2. Ethane (vehicular emission), 2-methylpentane (fuel evaporation), benzene (industry source), chloromethane (combustion), and ethyl acetate (solvent usage) were selected as tracers for five sources. Ethane concentration in Case 2 (5.9 ppbv) is much higher than in other processes, and ethane concentration during the recovery period (3.4 ppbv) is also higher than during the infection period (2.4 ppbv), which may to some extent reflect increased vehicular emissions during the recovery period.

Concentrations of most species were significantly higher during the recovery period than during the infection period. The representative pollution processes in both periods showed the same results as well, with a 79% higher concentration of TVOCs in Case 2 (65.1 ppbv) compared to Case 1 (36.3 ppbv) (Fig. 6). While in Case 1 industry was the dominant source of VOCs, by Case 2 motorized sources reached a concentration value of 21.2 ppbv, accounting for 33% of the observed VOCs, and became the dominant source of emissions. This is consistent with the fact that people's mobility activities have increased after the epidemic has entered the recovery period. As a group of VOCs species with the highest concentration share, ethane and propane contributed more to the clean days motor vehicle source than other processes, which also resulted in a 34% clean days motor vehicle source share.

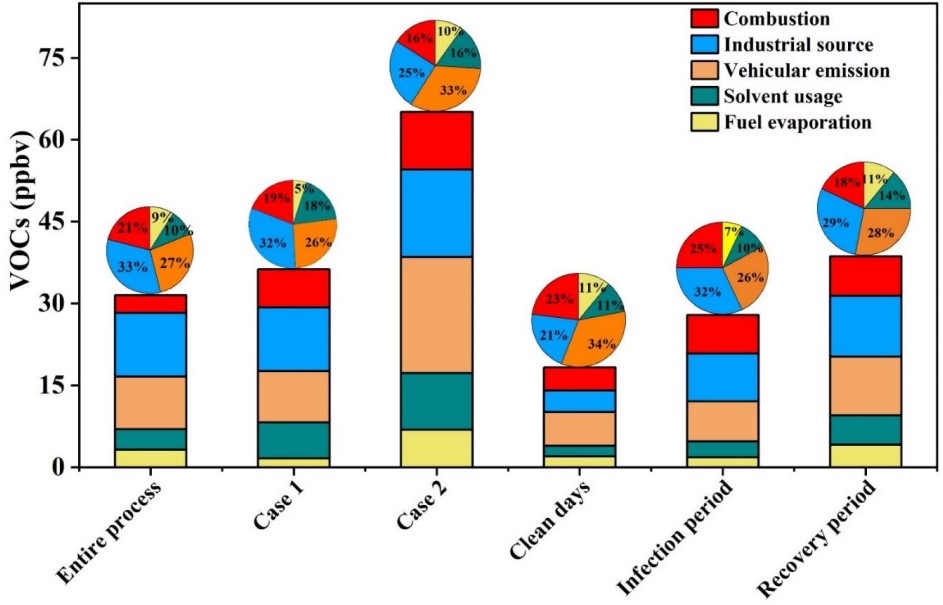

 Fig. 6. Contribution of each to VOCs for different processes.

## 3.3 SOAP

VOCs are estimated to contribute about 16−30% or more of PM$_{2.5}$ by mass through SOA production (Huang et al., 2014). Therefore, by calculating the SOAP value, the influence of different sources on PM$_{2.5}$ production can be reflected to a certain extent.

We have included quantitative analysis for SOAP as well. Fig. 7 shows the SOAP concentrations and contribution rates of the top ten species throughout the entire process, during two pollution processes, and clean days. The top ten species all reached close to 100% of the total SOAP contribution, with Case 1 reaching 98%. In each process, the composition of the top ten substances is essentially the same. Aromatic hydrocarbons contributed the most, with BTEX always occupying the top five positions and toluene the most. The SOAP values of the top ten contributing species for the two polluting processes are shown in Tables S3 and S4. Toluene, the highest contributing species, reached a SOAP value of 49.4 μg/m$^3$ in the most polluted Case 2, which was 3.2 times higher than the SOAP sum of all species on the clean day (15.5 μg/m$^3$). The SOAP value for Case 1, which is also a contaminated process, was 67 μg/m$^3$, and the main species (m/xylene: 9.8 μg/m$^3$, benzene: 8.5 μg/m$^3$) including toluene (34.6 μg/m$^3$) were lower than those for Case 2 (m/xylene: 19.4 μg/m$^3$, benzene: 13.4 μg/m$^3$).

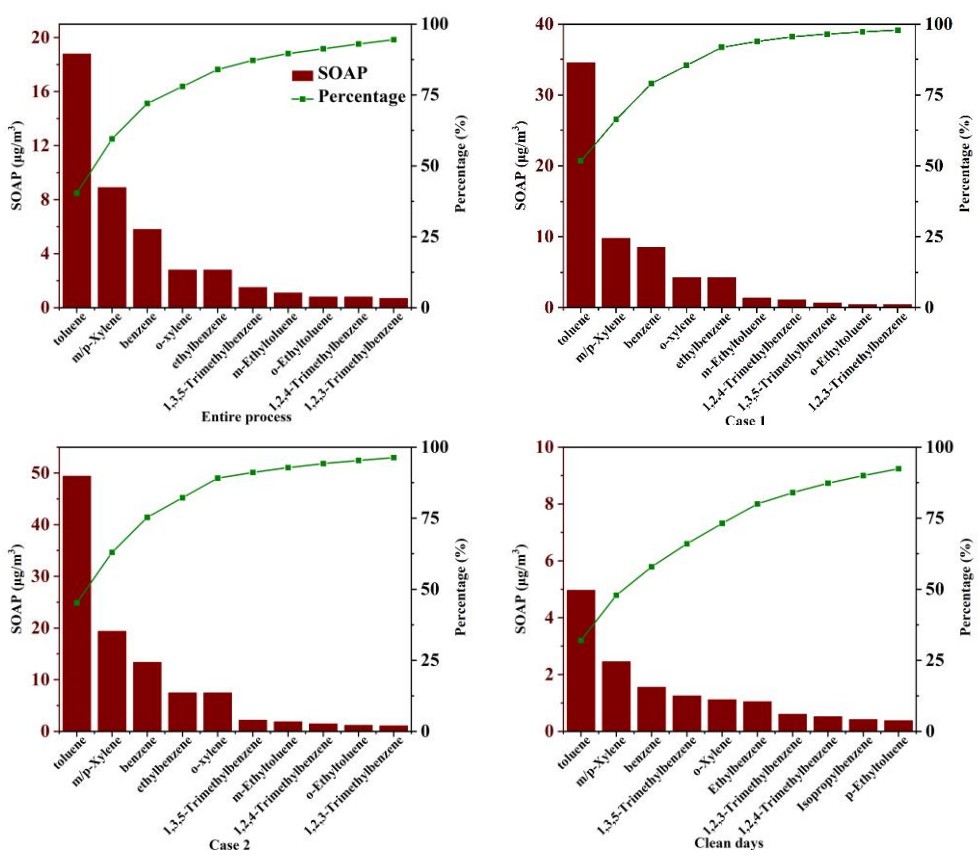

Fig. 7. SOAP dominant species in different processes

Figure 8 shows the SOAP calculated after source resolution of the two pollution

processes by PMF for clean days, respectively. In Case 1, industrial source is the
dominant source with a contribution ratio of 63%. In Case 2, the pollution sources
exhibit a more evenly distributed contribution, where the solvent usage and fuel
evaporation sources emerge as the primary contributors to SOAP, with their respective
contribution levels rising to 32% and 26%. Case 1 was during the infection period,
when social activities had not yet returned to normal. In Case 2, when society had
basically returned to normal, the increase in emissions from various sources resulted in
a more balanced distribution of SOAP contributions and caused more severe PM$_{2.5}$
pollution. In addition, a few days before Case 2, the Zhengzhou Municipal People's
Government initiated the Heavy Pollution Weather Level II response
(https://sthjj.zhengzhou.gov.cn/tzgg/7037130.jhtml) and introduced control measures
for emissions from industrial and mobile sources, which resulted in a significant
reduction of SOAP levels from industrial and motorized sources in Case 2. The clean
day result with a SOAP of 8.8 μg/m$^3$ also indicates that industrial and solvent usage
sources are the most dominant SOAP sources. The primary sources of aromatic
compounds, which are the most significant contributors to SOAP, are solvent usage and
industrial process emissions. This finding aligns with the results of other studies (Wu
et al., 2017). Consequently, it is imperative to implement measures to reduce PM$_{2.5}$
pollution by regulating emissions from industrial and solvent usage sources.

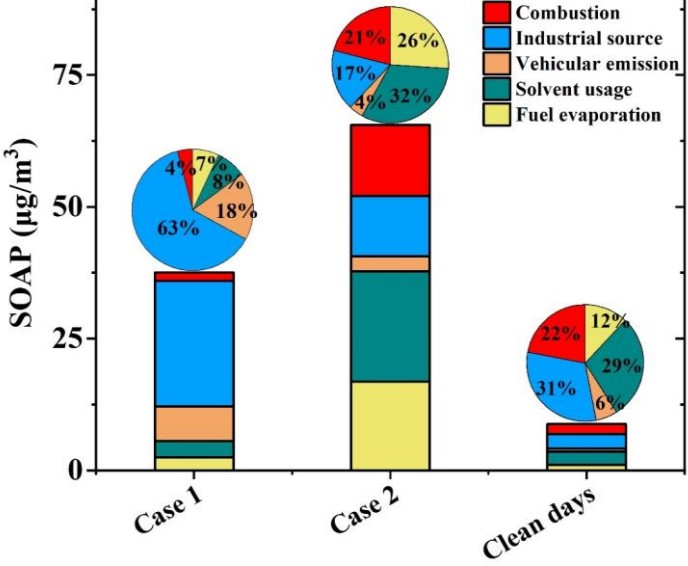


Fig. 8. SOAP value and contribution ratio of each process


## 4. Conclusions

Continuous observation of VOCs during the infection of the Omicron epidemic was carried out at an urban site in polluted Zhengzhou from December 1, 2022, to January 31, 2023. The daily average concentration of $PM_{2.5}$ ranged from 53.5 to 239.4 $\mu g/m^3$ with an average value of 111.5 ± 45.1 $\mu g/m^3$ during the whole period. The concentration of TVOCs ranged from 15.6 to 57.1 ppbv with an average of 36.1 ± 21.0 ppbv, higher than the same period in last year (27.9 ± 12.7 ppbv, Lai et al., 2024). Two representative contamination processes were identified (Case 1 during the infection period and Case 2 during the recovery period). While the meteorological conditions of the two pollution processes are relatively similar, the specific impacts caused thereby have yet to be determined. The concentration of TVOCs in Case 1 and Case 2 were 48.4 ± 20.4 and 67.6 ± 19.6 ppbv, respectively, increased by 63% and 188% compared with values during clean days. The average concentrations of $PM_{2.5}$ and TVOCs during Case 2 were 1.3 and 1.8 times of the values in Case 1. This is consistent with the observed increase in pollutant emissions following the return to normal social life from the period of Omicron infection. The highest volume contributions of alkanes were found both in Case 1 (48%) and Case 2 (44%). Though the volume contribution of aromatics were the lowest (6% in Case 1 and 7% in Case 2), the highest increase ratio was found from clean days to polluted episodes. Low wind speed and high humidity were the main meteorological reasons for the occurrence of pollution. Analyzing the sources of VOCs revealed that VOCs were found to be affected by a combination of local emissions and regional transport. The primary sources of atmospheric VOCs in Zhengzhou were identified as industrial emissions (32%), vehicle emissions (27%), and combustion (21%). Significant discrepancies were observed in the sources of VOCs between the two pollution processes. In Case 1, industrial emissions constituted the primary source of VOCs, accounting for 32% of the total VOC concentration. In contrast, in Case 2, the proportion of vehicle emissions increased to 33%, representing the primary source of VOCs.

A further analysis of the effect of VOCs on SOA generation reveals that aromatic compounds are the primary contributors to SOAP, with BTEX being the predominant contributor throughout the period. The SOAP values reached 37.6 and 65.6 $\mu g/m^3$ in Case 1 and Case 2, respectively. In Case 1, the greatest contribution to SOAP was made by industrial sources (63%, 23.8 $\mu g/m^3$), while vehicular sources, which constituted the

second most important source, accounted for only 18%. In Case 2, the contribution of each VOC source was more evenly distributed, with solvent use sources and fuel evaporation sources representing the primary contributors to SOAP, accounting for 32% (20.9 μg/m³) and 26% (16.8 μg/m³), respectively. The SOAP result for the clean day was 8.8 μg/m³, with industrial sources and solvent use still being the primary contributors. Therefore, the industrial and solvent use sectors are the predominant sources of pollutants during this observation. The aforementioned results substantiate the considerable impact of elevated emissions from all sources on the exacerbation of pollution following the conclusion of the Omicron infection.

**Author contribution:**

Bowen Zhang: Data curation, Methodology, Formal analysis, Writing Original Draft.

Dong Zhang: Data curation, Formal analysis, Review & Editing.

Zhe Dong: Data curation, Formal analysis, Review & Editing.

Xinshuai Song:  Data curation, Formal analysis.

Ruiqin Zhang: Supervision, Writing-Review & Editing, Funding acquisition.

Xiao Li: Formal analysis, Investigation, Supervision, Writing-Review & Editing.

**Competing interests:**

The contact author has declared that none of the authors has any competing interests.

**Acknowledgments:**

This research was supported by the Natural Science Foundation of Henan Province (232300421395) and the National Key Research and Development Program of China (2017YFC0212400).

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
