# Peer review of "The variations of VOCs based on the policy change of Omicron in traffic-hub city Zhengzhou"

_EGUsphere, 2024_

## Author Comment (AC1)

**Reviewer #1:**

We do appreciate your constructive and useful comments. To better reply to your general comments in your long paragraph, we have divided your comments into serval parts with superscript [a], [b], [c], etc., and correspondingly addressed your comments in a separate paragraph [a,b], etc. More detailed replies for the same topic are shown in your specific comments. To facilitate your review, the comments are in black, and the responses are in blue.

**Detailed comments:**

The article explored the relationship between VOCs and $PM_{2.5}$ with abundant VOCs species observed in Zhengzhou during the COVID-19 and made recommendations for the control of VOCs source emissions. [a]The current discussion may not be sufficiently supportive, please add more details to each section to make the entire article more logical.

[b]Basic details regarding instrumentation and data collection are missing. The authors need to supplement materials related to the reliability of the PMF results.

[c]Further more, more work is needed to elucidate the relationship between VOCs and haze pollution, as well as the influencing factors. And it is

suggested that model simulation on SOA formation potential be added to the manuscript.

While the theme and results of the study are interesting, I have provided a few suggestions for improvement.

Response: We are very grateful for the positive comments and suggestions. We have separately replied your suggestions into three parts as following:

[a]The current discussion may not be sufficiently supportive, please add more details to each section to make the entire article more logical.

We will overhaul every section of the revised version. In each chapter we will add more discussion to make the entire article more logical and comprehensive. Details can be found in the following point-to-point response.

[b]Basic details regarding instrumentation and data collection are missing. The authors need to supplement materials related to the reliability of the PMF results.

Additional details about the instruments and data collection are provided below.

For instrumentation comments, please see our replies in the following

specific comments.

Reliability of PMF results will be added to the text with relevant figures and tables in the supplementary materials.

We used displacement of factor elements (DISP) to assess PMF modelling uncertainty (for a description, see Paatero et al. (2014)). Q was less than 1% and no swaps occurred for the small est dQmax in DISP. Fpeak values from -2 to 2 were tested to explore the rotational stability of the solutions. $Q_{true}/Q_{exp}$ is lowest when Fpeak = 0, so we chose the PMF results for that case.

After examining 3-8 factors, 20 base runs with 5 factors eventually selected to represent the final result. We provide an explanation of factor selection in the supplementary materials. Figure 3(a) includes $Q_{true}/Q_{exp}$, $Q_{robust}/Q_{exp}$ for factors 3-8. The slopes of these two ratios in changed at five factors, and we found that five factors were more realistic after repeated comparisons of the results at four, five and six factors. These five factors eventually selected as potential sources for the observed VOCs are: (1) Fuel evaporation; (2) Solvent usage; (3) Vehicular emission; (4) Industrial source; and (5) Combustion.

References:

Paatero, P., Eberly, S., Brown, S. G., Norris, G. A.: Methods for

estimating uncertainty in factor analytic solutions, Atmospheric Measurement Techniques, Volume 7, 781-797, https:// 10.5194/amt-7-781-2014, 2014.

[c]Further more, more work is needed to elucidate the relationship between VOCs and haze pollution, as well as the influencing factors. And it is suggested that model simulation on SOA formation potential be added to the manuscript.

It is well known that VOCs are precursors for ozone formation and generation of secondary organic aerosols (SOAs). The $O_3$ pollution per se is not really a haze event. However, $O_3$ can assist the formation of fine particulates; there are numerous studies about the so-called double pollution of $O_3$ and $PM_{2.5}$. $O_3$ as an oxidant can improve the oxidation capacity and promote the oxidation of $SO_2$ and $NO_2$ (Li et al., 2023). On the other hand, the suppression of $O_3$ formation due to the presence of $PM_{2.5}$ has recently been highlighted for further $O_3$ pollution controls in regions that suffer high ozone concentrations (Zhang et al., 2024). Furthermore, $PM_{2.5}$ decreased the surface photolysis rates $J_{NO2}$ and $J_{O1D}$, resulting in a decrease in $O_3$ concentration in the VOC-sensitive area and a slight increase in the $NO_x$-sensitive area (Qu et al., 2023.). SOAs themselves are of course part of organic aerosols in $PM_{2.5}$ haze conditions.

The factors affecting VOC-haze interactions are typically atmospheric photochemistry and the mixing ratio of NOx and type of VOCs in generating SOAs. However, most *VOC species posed no non-carcinogenic risk during haze events (Zhang et al., 2021).*

Additionally, we have included quantitative analysis for SOA as well. In particular, Figure 1 shows the SOAP concentrations and contribution rates of the top ten species throughout the entire process, during two pollution processes, and clean days. The top ten species all reached close to 100% of the total SOAP contribution, with Case 1 reaching 98%. The composition of the top ten species is basically the same for each process. Toluene, m/p-xylene, and benzene were consistently the top three species. Toluene, the highest contributing species, reached a SOAP value of 49.4 $\mu g/m^3$ in the most polluted Case 2, which was 3.2 times higher than the SOAP sum of all species on the clean day (15.5 $\mu g/m^3$). The SOAP value for Case 1, which is also a contaminated process, was 67 $\mu g/m^3$, and the main species including toluene (34.6 $\mu g/m^3$) were lower than those for Case 2 (m/xylene: 9.8 $\mu g/m^3$, benzene: 8.5 $\mu g/m^3$) (m/xylene: 19.4 $\mu g/m^3$, benzene: 13.4 $\mu g/m^3$).

[Figure]

**Figure 1. SOAP dominant species in different processes**

The following is point-by-point responses to all your comments and valuable suggestions.

References:

Qu, Y.: The underlying mechanisms of PM$_{2.5}$ and O$_3$ synergistic pollution in East China: Photochemical and heterogeneous interactions, Science of The Total Environment, Volume 873, https://doi.org/10.1016/j.scitotenv.2023.162434, 2023.

Li, Y.: Spatiotemporal Variations of PM2.5 and O3 Relationship during 2014–2021 in Eastern China, Aerosol and Air Quality Research, https://doi.org/10.4209/aaqr.230060, 2023.

Zhang, D.: Characteristics, sources and health risks assessment of VOCs in Zhengzhou, China during haze pollution season, Journal of Environmental Sciences, Volume 108, https://doi.org/10.1016/j.jes.2021.01.035, 2021.

Zhang, J.: Enhanced summertime $PM_{2.5}$-suppression of $O_3$ formation over the Eastern U.S. following the $O_3$-sensitivity variations, Environmental Science: Atmospheres, 2024.

**1. Line 124-135:** The authors lack more detailed descriptions of the instrumentation. What are the working procedures of the instruments? What is the time resolution of the samples? How long were the samples collected for? Where were they captured? It is recommended to include information about instrument quality control methods.

Response: As per your comments, we have added a description of instrumental details including time resolution to the Materials and Methods section:

The VOCs were measured hourly using a GC-FID/MS (*TH-PKU 300 b*, Wuhan Tianhong Instruments Co., China). The instrument TH-PKU300b

includes electronic refrigeration ultra-low temperature pre-concentration sampling system, analysis system and system control software. The ambient VOCs in the first 5 minutes of each hour were collected by the sampling system and then entered the concentration system. Under low temperature conditions, the VOCs samples collected were frozen in the capillary capture column, and then quickly heated and resolved, so that the compounds entered the analysis system. After separation by chromatographic column, the compounds were monitored by FID and MS detectors. During the detection process, the atmospheric samples collected undergo analysis through two distinct pathways. C2-C5 hydrocarbons are analyzed using FID, while C5-C12 hydrocarbons, halocarbons, and OVOCs are analyzed with a MS detector. After excluding species with missing data exceeding 10%, the detected volatile organic compounds include 29 alkanes, 11 alkenes, 17 aromatics, 35halocarbons, 12 OVOCs, 1 alkyne (acetylene), and 1 sulfide ($CS_2$) with a total of 106 compounds.

As for information on instrument quality control methods, the revised text shall be:

The instrument was calibrated per week to ensure the accuracy of VOCs by injecting standard gases with a five-point calibration curve. The detection limit of C2-C5 hydrocarbons ranges from 0.007 to 0.099 ppbv,

other hydrocarbons are 0.004–0.045 ppbv, halogenated hydrocarbons 0.009-0.099 ppbv, OVOCs and other compounds of 0.006–0.095 ppbv. Thirty-two of the monitored VOCs had more than 90% of their data greater than the detection limit, and 34 had more than 50% of their data greater than the detection limit.

**2. Section 2.2 Positive Matrix Factorization (PMF) model**

How did the authors conduct factor selection, and why did not choose the 5-factor solution, 6-factor solution, and 7-factor solution? The authors need to provide more explanations and justifications in the manuscript.

Response: After examining 3-6 factors, 20 base runs with 5 factors eventually selected to represent the final result. We provide an explanation of factor selection in the Supplementary Materials. Figure 2(a) includes $Q_{true}/Q_{exp}$, $Q_{robust}/Q_{exp}$ for factors 3-8. The slopes of these two ratios in changed at five factors, and we found that five factors were more realistic after repeated comparisons of the results at four, five and six factors. These five factors eventually selected as potential sources for the observed VOCs are: (1) Fuel evaporation; (2) Solvent usage; (3) Vehicular emission; (4) Industrial source; and (5) Combustion. Five factors have been commonly reported before, e.g., in Shijiazhuang, northern China (Guan et al, 2023) and in Beijing (Cui et al., 2022). Figure 2(b) shows the result of Fpeak model run; $Q_{true}/Q_{exp}$ is lowest when Fpeak

= 0, so we chose the PMF results for that case.

The above statement will be incorporated into the revised text.

[Figure]

**Figure 2. (a) The Q$_{true}$/Q$_{expected}$ ratios in different solutions; (b) the Q$_{true}$/Q$_{expected}$ ratio for different Fpeak value solutions.**

References:

Cui, L., Wu, D., Wang, S., Xu, Q., Hu, R., and Hao, J.: Measurement report: Ambient volatile organic compound (VOC) pollution in urban Beijing: characteristics, sources, and implications for pollution control, Atmospheric Chemistry and Physics, 22, 11931-11944, https://doi.org/10.5194/acp-22-11931-2022, 2022.

Guan, Y., Liu, X., Zheng, Z., Dai, Y., Du, G., Han, J., Hou, L. a., and Duan, E.: Summer O3 pollution cycle characteristics and VOCs sources in a central city of Beijing-Tianjin-Hebei area, China,

Environmental Pollution, 323, 121293, https://doi.org/10.1016/j.envpol.2023.121293, 2023.

**3. Section 3.1 Pollution characteristics**

**Line 194:** Ensure that the font in the figures is consistently in Times New Roman. The y-axis labels do not match the legend (NO and $NO_x$).

Response: We have revised the manuscript according to your comments.

[Figure]

**Figure 3. Time series of WS, WD, T, RH, CO, PM$_{2.5}$, NO, TVOCs, NO$_x$ and O$_3$ during the observation period.**

4. What does the shading in Figure 1 represent? What are Case 1, Case 2,

Case 3, Case 4, and Case 5? Clear explanations need to be provided. If these cases represent haze pollution processes, how do you define your pollution processes? Please include the references you consulted.

Response: The shadow section in Figure 3 represents two haze pollution events during the monitoring period. A pollution event is determined when the daily average concentration of $PM_{2.5}$ exceeds 75 μg/m³ (China's II-level standard) for at least three consecutive days. We apologize for the unclear statement and recognize that the original annotations might confuse readers, so we simplify the labeling in Figure 1. To avoid misinterpretation, we deleted processes with no more than 3 days of continuous contamination in Figure 3. In the revised version, we focus on the distinct characteristics of Case 1, Case 2, and Clean days as depicted in the figure. Case 1 (December 5 to December 10 with daily average $PM_{2.5}$ = 142.5 μg/m³) and Case 2 (January 1 to January 8 with daily average $PM_{2.5}$ = 181.5 μg/m³) were selected as they represent the pollution events in infection and recovery periods, respectively, due to their long duration and high pollution levels. We divided this period into an infection period (1-30 December 2022) and a recovery period (1 January 2023-31 January 2023) based on Chinese Center for Disease Control and Prevention's December 2022-January 2023 infection data statistics (Figure 4). Any days with a $PM_{2.5}$ concentration lower than 35 μg/m³ (China's I-level standard) is considered as Clean days.

The above definition of pollution process will be incorporated into the revised manuscript.

[Figure]

**Figure 4. Trend of Omicron infection in China from 9 Dec. 2022 to 1 Jan. 2023 (CCDCP, 2023)**

**5. Line 217-225:** Why did you only discuss Case 1 and Case 3? Are these two periods particularly significant? Provide your reasoning.

Response**:** In this study, a continuous online observation of VOCs was carried out, which covered the abolishment of lockdown measures in Zhengzhou. A two-month-long lockdown measure was applied after first Omicron case of student in Zhengzhou University was confirmed on October 8, 2022. Lockdown measure was abolished from the beginning of December in 2022, which resulted in a sharp increase of Omicron-infected people and a decrease in daily social production activities. In fact, the "Nucleic Acid Screening Measures for all staff" policy was also

canceled on December 8 in 2022. People are basically homebound after the lifting of the lockdown policy due to either infection or fear of infection of Omicron variant. Due to herd immunization, people resumed normal life and industry normal activity. Therefore, the characteristics and variations of VOCs during different periods were investigated to assess their impact on pollution in general and on the formation of SOA in particular and to provide data support for future pollution control policies in Zhengzhou.

During the pollution events that occurred in the observation phase, Case 1 (December 5 to December 10) and Case 2 (January 1 to January 8) were considered to be in the early stages of infection and recovery periods, respectively. These two cases have long durations and high pollution levels, making them representative pollution processes for the infection and recovery periods. To avoid misinterpretation, we deleted processes with no more than 3 days of continuous contamination in Figure 1. Essentially, Case 3 in the original paper now is Case 2.

6. In Figure 2, the font should be changed to Times New Roman.

Response: We have modified in the revised manuscript according to your suggestions for the consistent font.

7. In this section, you only analyzed the variations in pollutant

concentrations and meteorological conditions. What is the relationship between them? Which factors are crucial causes of pollution? You have not provided analysis and explanations.

Response: The pollutant emission from different sources is the main cause of pollution. Indeed, meteorological conditions play an important role in the extent of pollution. But we know that the changes in emissions from pollution sources over a period of time are usually small, and meteorological conditions play a very important role in the formation of pollution. And previous studies have also shown that low wind speed, high relative humidity, and low precipitation are meteorological factors that contribute to the worsening of particulate matter pollution in Zhengzhou during winter (Duan et al., 2019). The meteorological conditions in the two periods are generally similar, and the Case 2 in the recovery periods are slightly more prone to atmospheric stability, high relative humidity and other meteorological conditions that are not conducive to the dispersion of pollutants than Case 1 in the infection periods. However, this slight meteorological difference cannot directly lead to a significant change in the degree of pollution we have observed. Clearly, the extent of pollution in different periods is mainly due to anthropogenic activities and to a lesser extent, regional transport (see the following reply), and not meteorological conditions. The reason for providing meteorological data is to add supplementary information for

these events.

Based on your comments, we have studied the relationship between meteorological conditions and the concentration of different pollutants. We found a significant correlation between relative humidity and the following three pollutants (Figure 5). It shows that changes in relative humidity have an important effect on pollution formation.

We will supplement this part according to your comments as: We analyzed the relationship between meteorological parameters and pollutant concentrations and found correlations between $PM_{2.5}$, TVOCs and $NO_x$ and RH, suggesting that meteorological conditions have an important influence on pollution formation.

[Figure]

**Figure 5. Relative humidity and (a) PM$_{2.5}$, (b) NO$_x$, (c) TVOCs correlation**

References:

Duan, S., Jiang, N., Yang, L., Zhang, R.: Transport Pathways and Potential Sources of PM$_{2.5}$ During the Winter in Zhengzhou, Environmental Science, Jan 8;40(1):86-93, https://doi.org/10.13227/j.hjkx.201805187, 2019.

8. Line 222-223: "[…] Among them, Case 1 (from December 5 to December 10 and […]" A closing bracket is missed.

Response: We have revised it in the manuscript.

**9. Section 3.2 Source appointment**

**Line 272:** 'indicating that the measured air VOC content was influenced by both remote sources and urban area emissions.', Are you referring to all VOCs? Or specifically to m/p-xylene and ethylbenzene?

Response: We apologize for the impact on your understanding due to our negligence. We are referring to m/p-xylene and ethylbenzene here.

**10. Line 271-273:** Your conclusion indicates that VOCs are influenced by transport and emissions from distant regions. Can this be further substantiated through transport or other means?

Response: We infer the photochemical age of the air mass by the ratio of X/E. When the ratio is significantly lower than 3, it indicates that VOC mainly migrates from long-distance sources (aged air masses) (Kumar et

al., 2018; Cerón Bretón et al., 2020). The average X/E value in this study was 2.0, indicating that the measured air VOCs content was affected by transport of nearby or long-distance source emissions.

To further confirm that VOCs are affected by long-range transport, we conducted a potential source analysis of VOCs.

The area covered by the airflow trajectory was gridded, and the 80th percentile values of TVOCs for each process were set as standard values to obtain a map of the potential source distribution of TVOCs. Areas with high PSCF values indicate potential source areas of VOCs pollution (Figure 6).

Figure 6 (a) shows the potential source analysis of VOCs during the infection period. The areas with the highest PSCF values (> 0.36, red) are found in Jincheng and Xi'an, northwest of Zhengzhou, and the areas with high PSCF values (> 0.28, orange) include Luoyang, Jiyuan, and north of Xuchang, which are all industrial-intensive cities. Figure 5 (b) shows the results of the recovery period, with a wider distribution of potential sources than the former, and a greater variation in the areas of high PSCF values. Compared with the previous month, Handan and Liaocheng areas become new high PSCF areas, the influence of Xi'an is weakened, and the yellow area (PSCF > 0.2) is shifted from the northwest to the northeast of Zhengzhou.

The above analysis can also show that the VOCs at the observation sites are mainly influenced by the transmission from the distant areas.

[Figure]

**Figure 6. Potential source areas for VOCs (a) Infection period (b) Recovery period (Black pentagrams represent sampling locations)**

Reference:

Cerón Bretón, J. G.: Health Risk Assessment of the Levels of BTEX in Ambient Air of One Urban Site Located in Leon, Guanajuato, Mexico during Two Climatic Seasons, Atmosphere, 11, 165, https://doi.org/10.3390/atmos11020165, 2020.

Kumar, A., Singh, D., Kumar, K., Singh, B. B., and Jain, V. K.:

Distribution of VOCs in urban and rural atmospheres of subtropical India: Temporal variation, source attribution, ratios, OFP and risk assessment, Science of the Total Environment, 613-614, 492-501, https://doi.org/10.1016/j.scitotenv.2017.09.096, 2018.

**11. Line 306:** 'olefins' should be corrected to 'alkenes'.

Response: We have modified it in the revised version.

**12. Line 316-325:** Have you performed PMF in Case 1, Case 3, and clean days? It is recommended to check whether the results of factor analysis are consistent in different conditions (Case 1, Case 2, and clean days) and compare the results.

Response: We have indeed performed PMF on infection period, recovery period, Case 1, Case 2 and clean days.

The PMF results for infection period (Dec 1 to 30, 2022), and recovery period (Jan 1 to 31), as well as the two pollution events and clean days, are shown in the figures below (Figure 7). They all exhibit the same 5 factors. It is worth noting that there are two y-axes in Figure 6: the left side represents the concentration of VOCs in units of ppbv, and the right side represents the percentage of specific VOCs within that factor.

Additionally, the concentration scales of some figures also differ.

Concentrations of most species were significantly higher during the recovery period than during the infection period. The representative pollution processes in both periods showed the same results as well, with a 79% higher concentration of TVOCs in Case 2 (65.1 ppbv) compared to Case 1 (36.3 ppbv) (Figure 8). While in Case 1 industry was the dominant source of VOCs, by Case 2 motorized sources reached a concentration value of 21.2 ppbv, accounting for 33% of the observed VOCs, and became the dominant source of emissions. This is consistent with the fact that people's mobility activities have increased after the epidemic has entered the recovery period. As a group of VOCs species with the highest concentration share, ethane and propane contributed more to the clean day motor vehicle sources than other processes, which also resulted in a 34% clean day motor vehicle source share.

[Figure]

**Case 1**

[Figure]

**Case 2**

[Figure]

**infection period**

[Figure]

**recovery period**

[Figure]

**Figure 7. Infection period, recovery period, high pollution events, and clean days PMF source analysis**

[Figure]

**Figure 8. Contribution of each source to VOCs for different processes**

**13. Section 3.3 SOAFP**

In this part, you only discuss the Case 1 and Case 2 processes, and you think that the control of $PM_{2.5}$ pollution in winter should focus on controlling vehicle emissions, solvent use, and combustion. I don't think it's convincing enough. It is recommended to add analysis of clean days. Contrast the pollution process with the clean day.

Response: VOCs are estimated to contribute about 16−30% or more of $PM_{2.5}$ by mass through SOA production (Huang et al., 2014). Therefore, by calculating the SOAP value, the influence of different sources on $PM_{2.5}$ production can be reflected to a certain extent.

We calculated the SOAP for the different processes from the PMF results in the previous question and added the results for the clean days as you suggested. The modified results are shown in Figure 9.

The SOAP of Case 2 was 65.6 μg/m³, which was much higher than that of Case 1 (37.6 μg/m³), and the main sources of SOAP differed significantly between the two pollution processes on the clean days. Industrial sources were absolutely dominant in Case 1 (63%). While in Case 2 the contribution of each pollution source is relatively more even, the

contribution of solvent use sources and fuel volatilization sources increases to 32% and 26% as the major SOAP sources. The result of clean day with SOAP of 8.8 μg/m³ also shows that industrial and solvent use sources are the most dominant SOAP sources. Therefore, there is a need to reduce $PM_{2.5}$ pollution by controlling emissions from industrial and solvent use sources.

[Figure]

**Figure 9. SOAP value and contribution ratio of each component**

References:

Huang, R. J.: High secondary aerosol contribution to particulate pollution during haze events in China, Nature 2014, 514 (7521), 218−22.

---

## Author Comment (AC2)

**Reviewer #2:**

We do appreciate your constructive and useful comments. To better reply to your overall comments in your long paragraph, we have divided your comments into serval parts with superscript [a], [b], [c], etc., and correspondingly addressed your comments in a separate paragraph [a, b], etc. More detailed replies are shown in your specific comments. To facilitate your review, the comments are in black, and the responses are in blue.

Overall comment:

The COVID-19 lockdown measures provide a natural experiment for probing air quality changes under substantial emission reductions. Zhang et al. investigate the variations of VOCs in response to the policy-driven emission changes in Zhengzhou city of China by using online ambient measurements, and the PMF model. [a]While this paper is within the scope of ACP, the present manuscript is limited to a cursory data analysis (simply reporting measurement results), without convincing evidence and in-depth discussion, which makes this paper unpublishable in the present form. [b]Further, the innovation of this work is far below the standard required to be published on ACP, which is even not qualified as a measurement report. [c]Though addressing the specific comments below may improve the paper, I don't think these improvements could justify publication in a high-standard journal such as ACP. Concerning the major

flaws and the lack of innovation, I think this paper should be rejected.

[a]While this paper is within the scope of ACP, the present manuscript is limited to a cursory data analysis (simply reporting measurement results), without convincing evidence and in-depth discussion, which makes this paper unpublishable in the present form.

We extend our heartfelt appreciation for your insightful comments. In response, we are committed to augmenting the manuscript with a more rigorous quantitative analysis and a profound exploration of the subject matter. Additionally, we undertook a comprehensive overhaul of the article to elevate its scholarly merit and overall quality.

We are sorry for the unclear and confusing statements in our original draft. Initially we had many cases (5 cases) in different studied periods exhibiting $PM_{2.5}$ pollution, and did not clearly explain why only discussing Case 1 and 3. We also did not clearly state the infection and recovery periods. These shortcomings including the annotations in Fig. 1 certainly confuse readers/reviewers.

Due to the lack of sufficient sampling days in other cases, we only discuss VOCs, and to a lesser extent, $PM_{2.5}$ changes in two major cases (Case 1 and 2 which is previous Case 3) along with clean days as well as infection and recovery periods; all due to the impact of ending China's

zero- COVID policy.

In the analysis section of the results discussion, we added quantitative analyses of the main VOC and SOAP species for the clean days and for the two pollution processes; in the PMF source analysis section we added CPF plots and in the supplementary Materials added plots of daily trends in the source analysis results, as well as the rationale for the selection of the PMF factors. The results of the infection period and the recovery period are also compared according to the updated VOCs source analysis results. In addition, the correlation analysis between meteorological conditions and pollutant concentrations, the analysis of potential pollution sources, the PMF factor profiles of different pollution processes, and the concentrations of the main tracers of different processes are added in the supplementary materials, which provide a more scientific basis for the conclusions in our manuscript.

[b]Further, the innovation of this work is far below the standard required to be published on ACP, which is even not qualified as a measurement report.

It is our fault not to clearly show the rationale for our study. This research investigation is centered on the examination of the fluctuations in VOCs and $PM_{2.5}$ pollution levels within Zhengzhou, following the relaxation of COVID-19 control measures with the emergence of COVID-19 variant.

While some atmospheric VOC studies involving the impact of Covid-19 lockdown have been performed in India (Singh et al., 2023a), in China (e.g., Pei et al., 2022; Jensen et al., 2023; Zuo et al., 2024), or with respect to BETX only (e.g., Sahu et al., 2022; Singh et al., 2023b), a gap persisted in the investigation of VOCs due to the impact of abolishment of China's zero-policy. Furthermore, the present study is focused on the period dominated by the COVID-19 Omicron variant, which exhibited distinct characteristics in terms of geographical spread, infected population size, and symptomatology compared to earlier strains (Petersen et al., 2022; Merino et al., 2023). This period also witnessed substantial alterations in China's pandemic zero-Covid control policy, resulting in significant changes in societal activities (Figure 1). Consequently, this study aims to a detailed examination of how the alteration influenced atmospheric pollution, particularly regarding VOCs.

[Figure]

**Figure 1. Trend of Omicron infection in China from 9 Dec. 2022 to 1**

**Jan. 2023 (CCDCP, 2023)**

The shadow section in Figure 2 represents two haze pollution events during the monitoring period. A pollution event is determined when the daily average concentration of $PM_{2.5}$ exceeds 75 μg/m$^3$ (China's II-level standard) for at least three consecutive days. We apologize for the unclear statement and recognize that the original annotations might confuse readers, so we simplify the labeling in Figure 2. To avoid misinterpretation, we deleted processes with no more than 3 days of continuous contamination in Figure 2. In the revised version, we focus on the distinct characteristics of Case 1, Case 2, and Clean days as depicted in the figure. Case 1 (December 5 to December 10 with daily average $PM_{2.5}$ = 142.5 μg/m$^3$) and Case 2 (January 1 to January 8 with daily average $PM_{2.5}$ = 181.5 μg/m$^3$) were selected as they represent the pollution events in infection and recovery periods, respectively, due to their long duration and high pollution levels. We divided this period into an infection period (1-30 December 2022) and a recovery period (1 January 2023-31 January 2023) based on Chinese Center for Disease Control and Prevention's December 2022-January 2023 infection data statistics (Figure 1). Any days with a $PM_{2.5}$ concentration lower than 35 μg/m$^3$ (China's I-level standard) is considered as Clean days.

[Figure]

**Figure 2. Time series of WS, WD, T, RH, CO, PM$_{2.5}$, NO, TVOCs, NO$_x$ and O$_3$ during the observation period.**

The above definition of pollution process will be incorporated into the revised manuscript.

References:

Jensen et al., 2023. Measurements of Volatile Organic Compounds During the COVID-19 Lockdown in Changzhou, China.

Pei et al., 2022. Decrease in ambient volatile organic compounds during the COVID-19 lockdown period in the Pearl River Delta region, south China.

Sahu et al., 2022. Impact of COVID-19 Pandemic Lockdown in Ambient Concentrations of Aromatic Volatile Organic Compounds in a Metropolitan City of Western India.

Singh et al., 2023a. Substantial Changes in Selected Volatile Organic Compounds (VOCs) and Associations with Health Risk Assessments in Industrial Areas during the COVID-19 Pandemic.

Singh et al., 2023b. Distribution and temporal variation of total volatile organic compounds concentrations associated with health risk in Punjab, India

Zuo et al., 2024. Pollution characteristics and source differences of VOCs before and after COVID-19 in Beijing

Merino et al., 2023. Evaluating the spread of Omicron COVID-19 variant in Spain

Petersen et al., 2022. Clinical characteristics of the Omicron variant - results from a Nationwide Symptoms Survey in the Faroe Islands, International Journal of Infectious Diseases,

[c]The major flaws and the lack of innovation.

The rationale for our study involved three major tasks: (1) Omicron variant; (2) abolishment of China's zero policy; and (3) detailed

VOC/PM$_{2.5}$ analysis. To our best knowledge this is the first attempt to evaluate the Omicron variant impact of ending China's zero-Covid policy on ambient VOCs and PM$_{2.5}$. We do hope that through refining and overhauling the article's content, we can deepen our analysis/discussion and potentially alter your negative view.

Please refer to the above brief rationale (innovation) for our study. We will try to explain the innovation of our work in more details below.

China lifted the zero-COVID strategies, notably by announcing the '10 measures' about the optimization of COVID-19 rules on 7 December 2022 (Xinhua, 2022). After that, China experiences a nationwide outbreak of COVID-19. Leung et al. (2023) estimated that the cumulative infection attack rate in Beijing was 75.7% (95% credible interval (CrI): 60.7–84.4) on 22 December 2022 and 92.3% (95% CrI: 91.4–93.1) on 31 January 2023. A recent study by Liang et al. (2023) showed that the cumulative SARS-CoV-2 infection rate rose rapidly to 70% within three weeks after the ending of the zero-COVID policy in Macao. A study conducted in Guangzhou also revealed that the infection attack ratio reached to 80.7% (95% CrI: 72.2–86.8) at 30 days after easing the zero-COVID policy (Huang et al., 2023)

Indeed, there have been some studies discussing the impact of human factors on air pollution during and after the outbreak of the Coronavirus

disease (e.g., Ma et al., 2022; Jiang et al., 2023; Song et al., 2023), but as mentioned earlier, only a few studies with in-depth exploration of the changes in VOCs and none dealing with ending the zero-Covid policy during Omicron variant infection period.

Our research primarily concentrates on the period dominated by COVID-19 Omicron variant, where they demonstrate notable differences from the early virus strains (i.e., original SARS-CoV-2 virus and Delta) in terms of geographical transmission, the scale of the infected population, and symptom manifestation.

The 7th announcement of 2022 issued by the National Health Commission of China states that, starting from January 8, 2023, the Class A infectious disease prevention and control measures specified in the Infectious Disease Prevention and Control Law of the People's Republic of China for COVID-19 will be lifted; COVID-19 will no longer be included in the quarantine infectious disease management stipulated by the Frontier Health and Quarantine Law of the People's Republic of China. This signifies a significant shift in China's pandemic control policy in comparison to the period preceding the issuance of the announcement. We believe that this change is worth exploring in terms of its impact on transportation and industrial production emissions. Essentially, this research serves to address the existing gap in the

literature concerning the effects of the Omicron variant on VOCs and PM$_{2.5}$ pollution levels in Zhengzhou amidst policy fluctuations, specifically the end of zero-Covid policy.

Our research findings also confirm that traffic emissions remain the primary source of pollution in Zhengzhou, thus providing valuable insights for formulating control measures.

References:

Huang, J.: Infection rate in Guangzhou after easing the zero-COVID policy: seroprevalence results to ORF8 antigen, The lancet Infectious Diseases, Volume 23, Issue 4, https://doi.org/10.1016/S1473-3099(23)00112-3, 2023.

Jiang, N., Hao, X., Hao, Q., Wei, Y., Zhang, Y., Lyu, Z., Zhang, R.: Changes in Secondary Inorganic Ions in PM$_{2.5}$ at Different Pollution Stages Before and After COVID-19 Control, Environmental Science, 0250-3301(2023)05-2430-11, https://doi.org/10.13227/j.hjkx.202206170, 2023.

Leung, K.: Estimating the transmission dynamics of SARS-CoV-2 Omicron BF.7 in Beijing after adjustment of the zero-COVID policy in

November–December 2022, Nature Medicine 29, 579–582, https://doi.org/10.1038/s41591-023-02212-y, 2023.

Liang, L.: Antibody drugs targeting SARS-CoV-2: Time for a rethink?, Biomedicine & Pharmacotherapy, Volume 176, https://doi.org/10.1016/j.biopha.2024.116900, 2024.

Ma, Q., Wang, W., Wu, Y., Wang, F., Jin, L., Song, Y., Han, Y., Zhang, R., Zhang, D.: Haze caused by $NO_x$ oxidation under restricted residential and industrial activities in a mega city in the south of North China Plain, Chemosphere, Volume 305, 135489, https://doi.org/10.1016/j.chemosphere.2022.135489, 2022.

Song, X., Zhang, D., Li, X., Lu, X., Wang, M., Zhang, B., Zhang, R.: Simultaneous observations of peroxyacetyl nitrate and ozone in Central China during static management of COVID-19: Regional transport and thermal decomposition, Atmospheric Research, Volume 294, 106958, https://doi.org/10.1016/j.atmosres.2023.106958, 2023.

Xinhua News Agency, The "new ten" to optimize the implementation of epidemic prevention and control is here, http://www.news.cn/politics/2022-12/07/c_1129189285.htm, 2022.

Major comments:

1) The major weakness of this work is the lack of innovation. The impacts of the Omicron outbreak on Chinese cities are already well-documented and extensive studies have been conducted to elucidate the role of the anthropogenic sector on air pollution during- and post-outbreak periods. The authors claimed that industrial and vehicular emissions are dominant sectors contributing to ambient VOC, which is quite clear in prior studies. Further, the changes in $PM_{2.5}$ and VOCs in response to the lockdown are broadly consistent with previous findings in Zhengzhou (even in Chinese literature). What is the innovation of this work and what are the new findings from this work that contribute to the air quality community?

Response: Again, we apologize for the lack of description of the rationale for our study and lack of in-depth analysis of our VOC results in our original draft. We have added the distribution of major flaws and the lack of innovation in the above comment, please see our point-by-point responses of ᶜThe major flaws and the lack of innovation.

 The usage of SOAP should be revisited. The authors should be aware that SOAP is a very simple metric that provides limited information on SOA formation potential because SOA yield for individual VOCs in China may vary significantly in other countries due to the different levels of NOx and other oxidants. SOAP is generally adopted to reflect the SOA

production potential based on bottom-up emission inventory (see Wu & Xie, ES&T), rather than using short-time ambient measurements. Therefore, I doubt the conclusion driven by the simple SOAP calculation. The authors should consider using F0AM and PBM-MCM for examining SOA production changes rather than SOAP.

Response: Thank you very much for your valuable advice. After carefully reading the literature you recommended, we found that the analysis conclusions about SOAP in Wu & Xie 's research have some similarities with ours (Wu et al., 2018). For example, Wu & Xie 's research found that aromatics contribute the most to SOAP, followed by alkanes and alkenes. Similarly, the results calculated using the toluene weighted mass contributions method (Derwent et al., 2010) also indicate that aromatics contribute the most to SOAP, followed by alkanes and alkenes. The toluene weighted mass contributions method has been widely used in calculating SOAP based on observed VOCs (Zhang et al., 2017; Hui et al., 2019; Li et al., 2020). Therefore, this method also has a certain representativeness. Of course, as you said, this is not the most appropriate method, and using F0AM and PBM-MCM for examining SOA production changes is a very good suggestion. However, due to the limitations of our related technologies, we are unable to use F0AM or PBM-MCM for examining SOA production changes. This is a very regrettable thing. Your suggestion has pointed out a very good direction

for our future research.

On the other hand, PBM-MCM can indeed effectively simulate atmospheric chemical processes in the troposphere under certain circumstances. Taking MCMv3.2 as an example, it includes 5900 species of reactants and 16500 chemical reactions. During the modeling process, a large number of model parameters need to be set, and it is influenced by various environmental variables such as temperature, atmospheric pressure, relative humidity, boundary layer height (Lam et al., 2013), among others. The–results may have significant errors compared to the true values. Furthermore, this model is often used to analyze the sources of atmospheric $O_3$ (Xie et al., 2021).

The issues faced when applying the F0AM model are similar as well. For example, the observed photodissociation frequency (J value) needs to be input. The photodissociation frequency controls the generation of free radicals and the lifetimes of many compounds. Due to the various influencing factors, the accurate simulation of J values is challenging. A major shortcoming of the modeling approach is the lack of explicit representation of transport processes (entrainment, dilution, etc.), which has several practical consequences. First, primary emissions like $NO_x$ and hydrocarbons must be constrained or otherwise re-supplied to compensate for chemical loss. Emissions can also be parameterized explicitly but

require knowledge of the boundary layer depth and assumed instantaneous mixing. Second, a generic "physical loss" lifetime of 6–48 h is often assigned to all species to mitigate build-up of long-lived oxidation products over multiple days of integration. Model users must be aware of the limitations imposed by these choices (Wolfe et al., 2016).

Even though the SOAP calculation process based on a coefficient of individual VOC species developed by Derwent et al. (2010) certainly has errors, it is our belief that SOA production obtained from F0AM and PBM-MCM models exhibited as many uncertainties as a simple Derwent's SOAP approach.

The importance of SOA to atmospheric problems is well known. Previous studies have used SOAP calculations to investigate the contribution of atmospheric VOCs to $PM_{2.5}$ production, demonstrating that contribution of different sources to the formation of SOA. (Shi et al., 2015; Liu et al., 2022; Liang et al., 2023). In our paper, SOAP values were determined to reflect the impact of the end of China's zero-Covid policy. But we will continue to work hard, hoping to include the analysis of F0AM and PBM-MCM in our future research.

References:

Derwent, R. G., Jenkin, M. E., Utembe, S. R., Shallcross, D. E.,

Murrells, T. P., and Passant, N. R.: Secondary organic aerosol formation from a large number of reactive man-made organic compounds, Science of the Total Environment, 408, 3374-3381, https://doi.org/10.1016/j.scitotenv.2010.04.013, 2010.

Hui, L., Liu, X., Tan, Q.: VOC characteristics, sources and contributions to SOA formation during haze events in Wuhan, Central China, Science of The Total Environment, Volume 650, Part 2, https://doi.org/10.1016/j.scitotenv.2018.10.029, 2019.

Lam, S.H.M., Saunders, S.M., Guo, H., Ling, Z., Jiang, F., Wang, X., Wang, T.: Modelling VOC source impacts on high ozone episode days observed at a mountain summit in Hong Kong under the influence ofmountain-valley breezes, Atmospheric Environment, 81, 166–176, https://doi.org/10.1016/j.atmosenv.2013.08.060, 2013.

Liang, S., Gao, S., Wang, S., Chai, W., Chen, W., Tang, G.: Characteristics, sources of volatile organic compounds, and their contributions to secondary air pollution during different periods in Beijing, China, Science of the Total Environment, Volume 858, Part 2, https://doi.org/10.1016/j.scitotenv.2022.159831, 2023.

Li, Q., Su, G., Li, C.: An investigation into the role of VOCs in SOA and ozone production in Beijing, China, Science of The Total

Environment, 720, https://doi.org/10.1016/j.scitotenv.2020.137536, 2020.

Liu, J., Chu, B., Jia, Y., Cao, Q., Zhang, H., Chen, T., Ma, Q.: Dramatic decrease of secondary organic aerosol formation potential in Beijing: Important contribution from reduction of coal combustion emission, Science of the Total Environment, 832, https://doi.org/10.1016/j.scitotenv.2022.155045, 2022.

Shi, J., Deng, H., Bai, Z., Kong, S., Wang, X., Hao, J., Han, X., and Ning, P.: Emission and profile characteristic of volatile organic compounds emitted from coke production, iron smelt, heating station and power plant in Liaoning Province, China, Science of the Total Environment, 515-516, 101-108, https://doi.org/10.1016/j.scitotenv.2015.02.034, 2015.

Wolfe, G.M., Marvin, M.R., Roberts, S.J., Travis, K.R., Liao, J.: The Framework for 0-D Atmospheric Modeling (F0AM) v3.1, Geoscientific Model Development, 9, 3309–3319, https://doi:10.5194/gmd-9-3309-2016, 2016.

Wu, R., Xie, S.: Spatial Distribution of Secondary Organic Aerosol Formation Potential in China Derived from Speciated Anthropogenic Volatile Organic Compound Emissions, Environmental Science &

Technology, 52, 8146-8156, https://doi:10.1021/acs.est.8b01269, 2018.

Xie, Y., Cheng, C., Wang, Z., Wang, K., Wang, Y., Zhang, X., Li, X., Ren, L., Liu, M., Li, M.: Exploration of $O_3$-precursor relationship and observation-oriented $O_3$ control strategies in a non-provincial capital city, southwestern China, Science of The Total Environment,800, 149422, https://doi.org/10.1016/j.scitotenv.2021.149422, 2021.

Zhang, Z., Wang, H., Chen, D.: Emission characteristics of volatile organic compounds and their secondary organic aerosol formation potentials from a petroleum refinery in Pearl River Delta, China, Science of The Total Environment, 584–585, 1162-1174, https://doi.org/10.1016/j.scitotenv.2017.01.179, 2017.

3) The captions of the results section are meaningless. "Pollution characteristics" and "Source appointment" is not clear to the readers and should be rewritten for clarification.

Response: We have modified "Pollution characteristics" to "Overview of variation in pollutants and meteorological parameters", and "Source appointment" to "Source Analysis of VOCs".

4) The writing of this paper is in need of much attention. Specifically, the writing suffers from a series of fundamental issues, including a lack of

clear organization, pervasive grammatical, and stylistic errors. I suggest the authors carefully read through the manuscript and rephrase the results section. There is substantial awkward phrasing throughout the paper which is confusing and misleading to the readers.

Response: We apologize for the grammatical and stylistic errors in the manuscript, and unclear statement which certainly confuses readers. We have addressed your comments in serval parts with superscript [a, b] above.

We will undertake extensive revisions and proofreading to enhance the clarity and coherence of the manuscript. This will ensure that the article is free from grammatical errors and stylistic issues, making it easier for readers to understand.

Minor comments:

1) The SOA formation potential is called "SOAP" rather than "SOAFP". The author should correct this abbreviation.

Response: After thorough examination of the literature, we found both usages in circulation. However, in the revised text, we shall adopt your suggested term of SOAP, as demonstrated in our replies to reviewers' comments.

---

## Author Comment (AC3)

**Reviewer #3:**

We do appreciate your constructive and useful comments. To better reply to your detailed comments in your long paragraph, we have divided your comments into serval parts with superscript a, b, c, and correspondingly addressed your comments in a separate paragraph a, b, etc. More detailed replies are shown in your specific comments. To facilitate your review, the comments are in black, and the responses are in blue.

Detailed comments:

To provide a fair assessment, I refrained from reading previous reviews of the manuscript. Zhang et al. investigated VOC emissions during winter in Zhengzhou, China, likely aiming to understand the impact of the Omicron lockdown on city pollutants. [a]However, given the extensive documentation of air quality studies during COVID pandemic and its variants, including Omicron, the manuscript lacks novelty in this context. [b]The discussion on source apportionment also falls short, lacking depth and quantitative analysis. [c]Overall, the manuscript does not meet the standards for publication in ACP. I recommend the authors revise the manuscript, enhance data analysis and interpretation, present their findings in a more scientifically rigorous manner, and plan to resubmit as a new submission.

Response: We first express gratitude for your encouragement of our paper revision for resubmittal. The following replies are for your general comments.

[a]However, given the extensive documentation of air quality studies during COVID pandemic and its variants, including Omicron, the manuscript lacks novelty in this context.

Response: It is undeniable that there have been numerous studies on air quality during the COVID-19 pandemic and its variants (including the Omicron variant). However, there is still a gap in the investigation of VOCs during the epidemic period in Zhengzhou; almost no VOC study before/after the impact of ending China's zero- COVID policy.

During the studied period, China experienced significant shifts in its control policies regarding the Omicron variant, which in turn caused substantial changes in social activities. Consequently, this study aims to delve into the impacts of the zero-COVID policy change on atmospheric pollution, with a particular focus on VOCs.

[b]The discussion on source apportionment also falls short, lacking depth and quantitative analysis.

Response: In the analysis section of the results discussion, we added quantitative analyses of the main VOC and SOAP species for the clean days and for the two pollution processes; in the PMF source analysis section we added CPF plots and in the supplementary Materials added plots of daily trends in the source analysis results, as well as the rationale for the selection of the PMF factors. The results of the infection period and the recovery period are also compared according to the updated VOCs source analysis results. In addition, the correlation analysis between meteorological conditions and pollutant concentrations, the analysis of potential pollution sources, the PMF factor profiles of different pollution processes, and the concentrations of the main tracers of different processes are added in the supplementary materials, which provide a more scientific basis for the conclusions in our manuscript.

[c]Overall, the ion in ACP. I recommend the authors revise the manuscript, enhance data analysis and interpretation, present their findings in a more scientifically rigorous manner, and plan to resubmit as a new submission.

Response: Your encouragement is greatly appreciated. We will overhaul the manuscript and plan to resubmit it.

The followings are our responses to all of your comments and valuable suggestions.

1、The manuscript lacks analysis of measurements during or post-Omicron period to provide relevant insights for policy and management. Authors have broadly compared two pollution cases with a clean day during the sampling period. Even in this regard, the poor labeling technique of Figure 1, makes it very confusing what are Cases 1 through 5. Discussions for Cases 2 and 4 are missing. I feel that the title is misleading as the data has not been leveraged to present relevant results related to the Omicron period and policy relevance.

Response: We are sorry for the unclear and confusing statements in our original draft. Initially we had many cases (5 cases) in different studied periods exhibiting $PM_{2.5}$ pollution, and did not clearly explain why only discussing Case 1 and 3. We also did not clearly state the infection and recovery periods. These shortcomings including the annotations in Fig. 1 certainly confuse readers/reviewers.

In our revised text, due to the lack of sufficient sampling days in other cases, we only discuss VOCs, and to a lesser extent, $PM_{2.5}$ changes in two major cases (Case 1 and 2 which is previous Case 3) along with clean days as well as infection and recovery periods; all due to the impact of ending China's zero- COVID policy.

We have also added a quantitative analysis of the dominant species of VOCs and SOAP during the Case 1 and Case 2.

China lifted the zero-COVID strategies, notably by announcing the '10 measures' about the optimization of COVID-19 rules on 7 December 2022 (Xinhua, 2022). After that, China experiences a nationwide outbreak of COVID-19. We divided this period into an infection period (1-30 December 2022) and a recovery period (1 January 2023-31 January 2023) based on Chinese Center for Disease Control and Prevention's December 2022-January 2023 infection data statistics (Figure 1). The data in Figure 1 shows that during the initial phase when the containment had just been lifted and Omicron was not widely spread, there were long periods of pollution (Case 1, December 5 to December 10, daily mean $PM_{2.5}$ = 142.5 $\mu g/m^3$). While during the peak of Omicron infections, there were several consecutive clean days. When the peak of Omicron infection ended and the recovery phase began, there was another prolonged period of pollution lasting 8 days (Case 2, January 1 to January 8 with daily average $PM_{2.5}$ = 181.5 µg/m³), which aligns with the actual situation of increased emission intensity due to intensified human activities. The aim of this data analysis is to confirm the correlation between the series of phenomena and the policies and Omicron infections.

[Figure]

**Figure 1. Trend of Omicron infection in China from 9 Dec. 2022 to 1 Jan. 2023 (CCDCP, 2023)**


[Figure]

**Figure 2. (a) The $Q_{true}/Q_{expected}$ ratios in different solutions; (b) the**
**$Q_{true}/Q_{expected}$ ratio for different Fpeak value solutions.**

[b]The use of median value to replace missing values is not a justifiable way to
treat the data, if the authors think so then needs to be discussed.

We reviewed the literature of relevant studies based on your suggestion and found that there have been previous studies that chose to use the median as a replacement for missing (Baudic et al., 2016). In addition, the EPA PMF 5.0 User Guide also recommends using the median as a proxy for missing values (Norris et al., 2014). Therefore, we believe this is a reasonable approach to the data.

[c]Authors should examine at least 100 base runs with different seed numbers to find the best solution.

The EPA PMF 5.0 User Guide recommends 20 base runs. We reviewed studies using the PMF model and found that many of the results were obtained from 20 base runs (Qu et al., 2018; Li et al., 2015). Therefore, the results obtained from 20 base runs are credible.


4、 While analyzing PMF factors, authors should use the time series trend, diurnal variations, use of wind speed and direction for identifying possible source sectors, and comparison with other inorganic tracers like trace gases to parameterize the PMF factors. Without some of these analyses, naming the factors just using the VOC profile may be inaccurate as there can be several sources for an individual VOC.

Response: Based on your suggestions, we have updated the PMF spectra and plotted the daily trends for the different sources and the CPF plots for each source.

[revised manuscript text omitted]

5、 The authors should analyze differences in PMF factors/source profiles during and post-Omicron lockdown days and between high pollution and clean days.

Response: The following additions are based on your comments. Figure 6 compares the differences in PMF factor/source profiles during the peak of Omicron infection with those during the recovery phase after the peak, as well as between contaminated and clean days.

The screening of observed VOC species and their inclusion into PMF model, followed by the application of the random seed approach for the examination of 20 baseline runs per process using 3-6 factors, resulted in the selection of 5 factors from the 20 baseline runs to represent the final results of 5 factors. These five factors included: (1) Fuel evaporation; (2) Solvent usage; (3) Vehicular emission; (4) Industrial source; and (5) Combustion (Figure 6). These 5 factors have been commonly reported before, e.g., in Shijiazhuang, northern China (Guan et al, 2023) and in Beijing (Cui et al., 2022). It is worth noting that there are two y-axes in Figure 6: the left side represents the concentration of VOCs in units of ppbv, and the right side represents the percentage of specific VOCs within that factor. Additionally, the concentration scales of some figures also differ. We present the concentrations of the five main VOCs in all five factors in Table 2. Ethane (vehicular emission), 2-methylpentane (fuel evaporation), benzene (industry source), chloromethane (combustion), and ethyl acetate (solvent usage) were selected as tracers for five sources.

Concentrations of most species were significantly higher during the recovery period than during the infection period. The representative pollution processes in both periods showed the same results as well, with a 79% higher concentration of TVOCs in Case 2 (65.1 ppbv) compared to Case 1 (36.3 ppbv) (Figure 7). While in Case 1 industry was the dominant source of VOCs, by Case 2 motorized sources reached a concentration value of 21.2 ppbv, accounting for 33% of the observed VOCs, and became the dominant source of emissions. This is consistent with the fact that people's mobility activities have increased after the epidemic has entered the recovery period. As a group of VOCs species with the highest concentration share, ethane and propane contributed more to the clean day motor vehicle sources than other processes, which also resulted in a 34% clean day motor vehicle source share.

It can be anticipated that certain sources may overlap, meaning that some VOCs emissions undoubtedly come from multiple sources. Taking ethane in Case 1 as an example, the largest source is vehicle exhaust emissions (2.55 ppbv, 30%), followed by industrial emissions (2.54 ppbv, 30%), combustion sources (1.80 ppbv, 21%), solvent usage (1.32 ppbv, 16%), and fuel
evaporation (0.19 ppbv, 2%). The total was 8.4 ppbv, which is somewhat
different from the observed values (Table 2). At the same time, there are
cases where the observed values are perfectly matched, e.g., for 2-
methylpentane in the whole process. Similarly, this discrepancy is due to the
simple fact that the PMF model cannot fully explain the observed values at
100%.

[Figure]

**recovery period**

[Figure]

**infection period**

[Figure]

[Figure]

**Case 2**

[Figure]

**Clean Days**

**Figure 6. Infection period, recovery period, high pollution events, and clean days PMF source analysis**

[Figure]

**Figure 7. Contribution of each source to VOCs for different processes**

**Table 2. Concentrations of important tracer substances in different processes (ppbv) (observations in parentheses, red text indicates the corresponding source concentration of the substance)**

| Source | ethane | | | | | | 2-Methylpentane | | | | | |
|---|---|---|---|---|---|---|---|---|---|---|---|---|
| | Infection | Recovery | Entire | Case 1 | Case 2 | Clean | Infection | Recovery | Entire | Case 1 | Case 2 | Clean |
| **Factor 1** **Fuel evaporation** | 0.09 | 0.73 | 0.41 | 0.19 | 0.55 | 0 | **0.09** | **0.12** | **0.10** | **0.12** | **0.13** | **0.08** |
| **Factor 2** **Solvent usage** | 0.14 | 0.30 | 0 | 1.32 | 1.38 | 0.34 | 0.01 | 0.01 | 0.01 | 0.16 | 0 | 0 |
| **Factor 3** **Vehicle emission** | **2.39** | **3.35** | **2.91** | **2.55** | **5.85** | **2.12** | 0.02 | 0.06 | 0.06 | 0.03 | 0.16 | 0.02 |
| **Factor 4** **Industrial source** | 1.83 | 2.77 | 2.5 | 2.54 | 3.84 | 0.85 | 0.06 | 0.07 | 0.07 | 0.01 | 0.05 | 0.03 |
| **Factor 5** **Combustion** | 1.55 | 0.76 | 1.36 | 1.80 | 0.43 | 1.17 | 0.04 | 0.02 | 0 | 0 | 0.10 | 0 |
| **sum** | **6.00** **(6.80)** | **7.91 (7.81)** | **7.18** **(6.80)** | **8.40** **(10.06)** | **12.05** **(12.17)** | **4.48** **(4.30)** | **0.22** **(0.25)** | **0.28** **(0.26)** | **0.24** **(0.24)** | **0.32** **(0.37)** | **0.44** **(0.45)** | **0.13** **(0.14)** |
| | benzene | | | | | | methyl chloride | | | | | |
| **Factor 1** | 0.02 | 0 | 0.06 | 0.04 | 0.01 | 0.06 | 0.02 | 0 | 0.08 | 0.05 | 0.14 | 0.07 |

| | | | | | | | | | | | | |
|---|---|---|---|---|---|---|---|---|---|---|---|---|
| **Fuel evaporation** | | | | | | | | | | | | |
| **Factor 2** Solvent usage | 0.13 | 0.26 | 0.16 | 0.17 | 0.57 | 0 | 0.18 | 0.09 | 0 | 0.23 | 0 | 0.04 |
| **Factor 3** Vehicle emission | 0.01 | 0.03 | 0.07 | 0.15 | 0.15 | 0.01 | 0.06 | 0.23 | 0.06 | 0.07 | 0.34 | 0.12 |
| **Factor 4** **Industrial source** | **0.16** | **0.19** | **0.09** | **0.36** | **0.63** | **0.06** | 0 | 0.13 | 0.30 | 0.27 | 0.11 | 0 |
| **Factor 5** **Combustion** | 0.24 | 0.3 | 0.33 | 0.16 | 0.31 | 0.08 | **0.58** | **0.91** | **0.72** | **0.55** | **1.67** | **0.35** |
| **sum** | **0.56 (0.65)** | **0.78 (0.83)** | **0.71 (0.69)** | **0.88 (1.10)** | **1.67 (1.74)** | **0.21 (0.20)** | **0.84 (0.99)** | **1.36 (1.43)** | **1.16 (1.14)** | **1.17 (1.37)** | **2.26 (2.35)** | **0.58 (0.54)** |

| | **ethyl acetate** | | | | | |
|---|---|---|---|---|---|---|
| | **Infection** | **Recovery** | **Entire** | **Case 1** | **Case 2** | **Clean** |
| **Factor 1** Fuel evaporation | 0 | 0 | 0.01 | 0.02 | 0.03 | 0 |
| **Factor 2** **Solvent usage** | **0.27** | **0.27** | **0.72** | **0.63** | **0.80** | **0.02** |
| **Factor 3** Vehicle emission | 0.08 | 0.01 | 0.03 | 0.01 | 0 | 0.01 |
| **Factor 4** Industrial source | 0 | 0 | 0.02 | 0.08 | 0.16 | 0.01 |

| | | | | | | | | | | | |
|---|---|---|---|---|---|---|---|---|---|---|---|
| **Factor 5** **Combustion** | 0 | 0.06 | 0.01 | 0.01 | 0 | 0.04 | | | | | |
| **sum** | **0.35 (0.45)** | **0.34 (0.40)** | **0.79 (0.68)** | **0.75 (0.81)** | **0.99 (1.09)** | **0.08 (0.06)** | | | | | |

---

## Author Response (AR2)

Title: Simultaneous observations of peroxyacetyl nitrate and ozone in central China during static management of COVID-19: Regional transport and thermal decomposition.

Authors: Bowen Zhang[1,3], Dong Zhang[2,3], Zhe Dong[2,3], Xinshuai Song[1,3], Ruiqin Zhang[1,3], Xiao Li[1,3,*]

Manuscript number: egusphere-2024-575

**Dear Editor**

Thank you for recognizing our work and encouraging us to resubmit the above manuscript. We would also like to thank the reviewers for reading our manuscript and again providing valuable comments and suggestions to improve the quality of our manuscript. We believe that all of the reviewers' comments have been addressed and we respond to each comment individually below. In order to incorporate the reviewers' comments into the revised manuscript, we will definitely revise the manuscript completely and analyze the data in more depth. We have also made changes to the abstract to highlight the substantive and novel contributions of the paper. Changes made in response to these responses are highlighted in yellow in the highlighted copy of the revised version. Our own minor changes are highlighted in red.

The following is a point-by-point response to each reviewer's comments.

**Reviewer #1:**

The authors have made significant efforts to address the comments and suggestions from the previous review. The revisions have improved the manuscript, but there are still some issues that need to be addressed before it can be considered for publication.

We do appreciate your recognition of our work. We would like to thank you again for your valuable suggestions. Your comments will still receive point-by-point responses.

Detailed comments:

Line 42: The current list of keywords is quite limited. I recommend including additional relevant terms that reflect the core topics and methodologies discussed in the manuscript.

Response: We are very grateful for the positive comments and suggestions. We have added 'Positive Matrix Factorization model' and 'Secondary organic aerosol formation potential' as new keywords to the manuscript.

Section 3.1 Overview of variation in pollutants and meteorological parameters

While the correlations between $PM_{2.5}$, TVOCs, $NO_x$, and relative humidity are mentioned, a more detailed analysis of these correlations is needed.

Explain the nature of these relationships and their impact on understanding pollution formation, and compare these correlations with previous research results. Although the influence of meteorological conditions on pollution formation is noted, further elaboration on how specific meteorological factors (such as low wind speed and temperature) affect pollutant concentrations is necessary. Discuss why these conditions might lead to higher or lower pollutant levels. Additionally, while the comparison of pollutant concentrations between different periods is addressed, a detailed analysis of the observed trends, such as the increase in $PM_{2.5}$ and TVOCs during the recovery period, should be provided. Analyze the possible reasons for these trends and their relationship with the resumption of production activities.

Response: We are very grateful for the positive comments and suggestions.

Emissions of pollutants from different sources are the main cause of pollution. It is true that meteorological conditions play an important role in the level of pollution. However, we know that emissions from pollutant sources usually change very little over a period of time, while meteorological conditions play a very important role in the formation of pollution. And previous studies have shown that meteorological factors such as low wind speed, high relative humidity, and low precipitation are responsible for the increase in particulate matter pollution in Zhengzhou in

winter (Duan et al., 2019). The meteorological conditions of the two time periods are basically similar, and Case 2 during the recovery period is slightly more prone to meteorological conditions unfavorable to pollutant dispersion, such as atmospheric stability and high relative humidity, than Case 1 during the infection period. However, this minor meteorological difference does not directly lead to significant changes in the pollution levels we observe. It is clear that pollution levels over time are primarily the result of anthropogenic activities and, to a lesser extent, regional transport (see responses below), rather than meteorological conditions. The reason for providing meteorological data is to add supplementary information to these events.

Our correlation analysis of different pollutants with meteorological conditions during pollution revealed that $PM_{2.5}$, TVOCs, and $NO_x$ were all positively correlated with relative humidity, which is consistent with the results of some previous studies (Wang et al., 2019). The high humidity environment favors the conversion of gaseous pollutants such as sulfur dioxide, nitrogen oxides, and ammonia into particulate matter, and the formation of static weather under meteorological conditions such as low wind speed, high humidity, and temperature inversion is the main factor for the occurrence of heavy pollution days.

We added the meteorological parameters of clean days in Table 1, and after

comparing the polluted days with the clean days, it can be seen that the wind speed is lower than that of the clean days, and the humidity is higher than that of the clean days, which is in line with the meteorological conditions characterized by the emergence of polluted days. However, when comparing the meteorological conditions of the two pollution processes, none of the processes showed a tendency to be more prone to pollution. However, the pollution parameters were significantly higher in Case 2 than in Case 1, a trend that is most likely related to the resumption of production activities and the increase in emissions during the Case 2 period.

Reference:

Duan, S., Jiang, N., Yang, L., Zhang, R.: Transport Pathways and Potential Sources of $PM_{2.5}$ During the Winter in Zhengzhou, Environmental Science, Jan 8;40(1):86-93, https://doi.org/10.13227/j.hjkx.201805187, 2019.

Wang, H., Li, J., Peng, Y., Zhang, M., Che, H., Zhang, X.: The impacts of the meteorology features on PM2.5 levels during a severe haze episode in central-east China, Atmospheric Environment, Volume 197, Pages 177-189, ISSN 1352-2310, https://doi.org/10.1016/j.atmosenv.2018.10.001, 2019.

Section 3.2 SOAFP

The analysis of SOAP contributions across different pollution processes provides an overview but lacks in-depth explanations. It would be beneficial to include more detailed analysis on why the industrial source is dominant in Case 1 and why solvent usage and fuel evaporation sources are more evenly distributed in Case 2. Additionally, analyze whether the

observed trends in SOAP values and source contributions are related to other environmental factors or changes over time. Explain how these trends impact air quality and $PM_{2.5}$ pollution.

Response: We are very grateful for the positive comments and suggestions. Case 1 was during the infection period, when social activities had not yet returned to normal. In Case 2, when society had basically returned to normal, the increase in emissions from various sources resulted in a more balanced distribution of SOAP contributions and caused more severe $PM_{2.5}$ pollution. In addition, a few days before Case 2, the Zhengzhou Municipal People's Government initiated the Heavy Pollution Weather Level II response (https://sthjj.zhengzhou.gov.cn/tzgg/7037130.jhtml) and introduced control measures for emissions from industrial and mobile sources, which resulted in a significant reduction of SOAP levels from industrial and motorized sources in Case 2.

Section 4 Conclusions

The conclusions provided primarily summarize the results without offering detailed explanations for the observed changes or the influence of different sources on VOCs and $PM_{2.5}$ pollution. To strengthen the conclusions, it would be beneficial to include more in-depth analysis and discussion regarding why certain changes were observed and how they relate to the influence of specific sources on VOCs and $PM_{2.5}$ formation, how different

sources contribute to changes in TVOCs and $PM_{2.5}$ levels across different periods, and potential driving factors behind the presence and variation of these sources during different pollution episodes. Incorporating such explanations will provide a clearer understanding of the underlying mechanisms driving the observed results and enhance the overall impact of the conclusions.

Response: We are grateful for your insightful feedback and have revised the abstract in accordance with your recommendations. The revised conclusion is presented below:

From December 1, 2022, to January 31, 2023, continuous observations of VOCs were conducted in a heavily polluted urban area of Zhengzhou during the Omicron epidemic infection. During the aforementioned period, the daily mean concentrations of $PM_{2.5}$ exhibited a range of 53.5 to 239.4 µg/m³, with a mean value of 111.5 ± 45.1 µg/m³. The concentrations of TVOCs ranged from 15.6 to 57.1 ppbv, with a mean value of 36.1 ± 21.0 ppbv, which was higher than that observed during the same period in the previous year (27.9 ± 12.7 ppbv, as reported by Lai et al., 2024). Two representative pollution processes were identified during the observation period: Case 1, which occurred during the infection period, and Case 2, which occurred during the recovery period. The TVOC concentrations in Case 1 and Case 2 were 48.4 ± 20.4 and 67.6 ± 19.6 ppbv, respectively,

which represented a 63% and 188% increase compared to the concentrations observed on clean days. The mean $PM_{2.5}$ and TVOC concentrations in Case 2 were 1.3 and 1.8 times higher, respectively, than those in Case 1. This is consistent with the observed increase in pollutant emissions following the return to normal social life from the period of Omicron infection. The volume contribution of alkanes is the highest in Case 1 (48%) and Case 2 (44%). Despite aromatic hydrocarbons exhibiting the lowest volumetric contribution (6% in Case 1 and 7% in Case 2), the greatest rate of increase in the volumetric contribution of aromatic hydrocarbons was observed from the clean day to the contaminated day. Low wind speed and high humidity were the main meteorological reasons for the occurrence of pollution. Analyzing the sources of VOCs revealed that VOCs were found to be affected by a combination of local emissions and regional transport. The primary sources of atmospheric VOCs in Zhengzhou were identified as industrial emissions (32%), vehicle emissions (27%), and combustion (21%). Significant discrepancies were observed in the sources of VOCs between the two pollution processes. In Case 1, industrial emissions constituted the primary source of VOCs, accounting for 32% of the total VOC concentration. In contrast, in Case 2, the proportion of vehicle emissions increased to 33%, representing the primary source of VOCs.

A further analysis of the effect of VOCs on SOA generation reveals that

aromatic compounds are the primary contributors to SOAP, with BTEX being the predominant contributor throughout the period. The SOAP values reached 37.6 and 65.6 μg/m³ in Case 1 and Case 2, respectively. In Case 1, the greatest contribution to SOAP was made by industrial sources (63%, 23.8 μg/m³), while vehicular sources, which constituted the second most important source, accounted for only 18%. In Case 2, the contribution of each VOC source was more evenly distributed, with solvent use sources and fuel evaporation sources representing the primary contributors to SOAP, accounting for 32% (20.9 μg/m³) and 26% (16.8 μg/m³), respectively. The SOAP result for the clean day was 8.8 μg/m³, with industrial sources and solvent use still being the primary contributors. Therefore, the industrial and solvent use sectors are the predominant sources of pollutants during this observation. The aforementioned results substantiate the considerable impact of elevated emissions from all sources on the exacerbation of pollution following the conclusion of the Omicron infection.

**Reviewer #2:**

I would like to thank the authors for submitting the review of your manuscript. However, I regret to recommend not publishing the manuscript in its current form as it still lacks clarity and scientific significance on how the results have been presented. Below, I have outlined only some of my specific concerns.

I am grateful for your time and effort in reviewing our paper and for your numerous valuable suggestions. We regret any shortcomings in the manuscript that did not meet your expectations and are committed to further revisions. We will provide a point-by-point response to your suggestions below.

Abstract:

1. Line 16: The abrupt mention of Case 2 is confusing. How many cases were analyzed, and based on what factors were these cases chosen?

Response:

1. We extend our sincerest apologies for any confusion that may have arisen from the reading material. In light of your invaluable suggestions, we have made the necessary additions to the summary.

2. Lines 23-25: The significance of this section is unclear. Please clarify what you mean and what you are trying to signify.

3. The abstract needs a more cohesive approach to presenting the results.

2 & 3. In accordance with your recommendations, we have implemented comprehensive revisions to the summary. The following revisions have been made:

Line10-39: Online volatile organic compounds (VOCs) were monitored before and after the Omicron policy change at an urban site in polluted Zhengzhou from December 1, 2022, to January 31, 2023. The characteristics and sources of VOCs were investigated. The daily mean concentrations of $PM_{2.5}$ and total VOCs (TVOCs) ranged from 53.5 to 239.4 µg/m³ and 15.6 to 57.1 ppbv, respectively, with mean values of 111.5 ± 45.1 µg/m³ and 36.1 ± 21.0 ppbv, respectively, throughout the period. Two severe pollution events (designated as Case 1 and Case 2) were identified in accordance with the National Ambient Air Quality Standards (NAAQS) (China's National Ambient Air Quality Standards (NAAQS) from 2012). Case 1 (December 5 to December 10, $PM_{2.5}$ daily mean = 142.5 µg/m³) and Case 2 (January 1 to January 8, $PM_{2.5}$ daily mean = 181.5 µg/m³) occurred during the infection period (when the policy of "full nucleic acid screening measures" was in effect) and the recovery period (after the policy was cancelled), respectively. The $PM_{2.5}$ and TVOCs values for Case 2 are, respectively, 1.3 and 1.8 times higher than those for Case 1. The results of the positive matrix factor modeling demonstrated that the

primary source of volatile organic compounds (VOCs) during the observation period was industrial emissions, which constituted 32% of the total VOCs, followed by vehicle emissions (27%) and combustion (21%). In Case 1, industrial emissions constituted the primary source of VOCs, accounting for 32% of the total VOCs. In contrast, in Case 2, the contribution of vehicular emission sources increased to 33% and became the primary source of VOCs. The secondary organic aerosol formation potential for Case 1 and Case 2 were found to be 37.6 $\mu g/m^3$ and 65.6 $\mu g/m^3$, respectively. In Case 1, the largest contribution of SOAP from industrial sources accounted for the majority (63%, 23.8 $\mu g/m^3$), followed by vehicular sources (18%). After the end of the epidemic and the resumption of productive activities in the society, the difference in the proportion of SOA generated from various sources decreased. Most of the SOAP came from solvent use and fuel evaporation sources, accounting for 32% (20.9 $\mu g/m^3$) and 26% (16.8 $\mu g/m^3$), respectively. On days with minimal pollution, industrial sources and solvent use remain the main contributors to SOA formation. Therefore, regulation of emissions from industry, solvent-using industries and motor vehicles need to be prioritized to control the $PM_{2.5}$ pollution problem.

Introduction:

1. Line 97: The term "human factor" is vague and hard to understand. Please specify what you mean.

Response:

1. Thank you for your suggestion. We have revised the sentence to "Furthermore, some studies have discussed the impact of changes in human production activities on air pollution during and after the outbreak of the coronavirus disease."

2. Lines 120-125: The phrase "See what results show later on..." is unclear. What results are you referring to, and how do they relate to the study's objectives?

2. We appreciate your input and regret that we are unable to provide a satisfactory response. A thorough examination of the section in question revealed no evidence of the suggested sentence in the manuscript. Should you believe that a problem persists, we kindly request that you contact us at your earliest convenience so that we may undertake a careful revision in accordance with your suggestions.

3. Line 110: The rationale for choosing this sampling duration is unclear. Was the Omicron lockdown a significant interruption compared to earlier variants? Providing a timeline of Omicron detection, lockdown periods, and sampling coverage would help clarify this. Additionally, what is meant by "nuclei acid screening measure for all staff," and how does it relate to your study? Clarify whether "all staff" refers to government staff or the general population.

3. Zhengzhou Municipal Government in Henan Provincial People's Government Portal (www.henan.gov.cn) on October 5, 2022 issued Circular No. 139, the content of the circular is due to Zhengzhou during this period Omicron infection cases frequently, in order to prevent the spread of the epidemic hidden, and therefore to carry out the city's new coronavirus nucleic acid screening, screening scope for all residents in the city area, at the same time, closed public places Suspension of business, and to advise residents to reduce activities outside, and the content of the above notice similar to the notice continued to be issued until the Circular No. 162. The epidemic prevention and control measures in Zhengzhou changed to tin Circular No. 163 issued on December 4, 2022, restoring the opening of closed public places; Circular No. 164 issued on December 8 announced that it was no longer necessary to present health codes and nucleic acid negative certificates for the movement of people, and that centralized isolation would no longer be adopted for those who were positive for the infection. Since then, the number of people moving around Zhengzhou has increased and social production has resumed.

After the quarantine policy was lifted, people basically rested at home due to infection or fear of infection with Omicron. The resumption of normal production and life depends on herd immunization. This outbreak event is the longest in duration and the largest in number of infections since the 2020 outbreak of the novel coronavirus in Zhengzhou. It would be

beneficial to investigate the impact of this event on emissions related to transportation and industrial production.

4. The introduction should clearly explain Cases 1 and 2, as well as the clean period, to clarify the study's objectives. Although some of this information will appear in the methods section, the introduction and abstract should provide a clear understanding of the study's purpose.

4. Thanks for your suggestion, we have added this section of the introduction to the narrative.

Line 105-109: The focus of this study was on pollution events in which the daily average $PM_{2.5}$ concentration exceeded 75 μg/m³ (China's Class II standard) for more than three consecutive days, and any day in which the $PM_{2.5}$ concentration was less than 35 μg/m³ (China's Class I standard) was considered a clean day.

Methods:

1. Line 140: The claim that the sampling period covered the entire infection period of Omicron is questionable. According to your introduction, Omicron started on October 8th and lasted until early December, while your sampling is from December 1st to January 31st. Your sampling period appears to coincide with the post-Omicron period.

Response:

1. The Zhengzhou Municipal Government issued Circular No. 139 on the Henan Provincial People's Government Portal (www.henan.gov. cn) on October 5, 2022, which stated that due to the frequent occurrence of omicron infections in Zhengzhou during this period, in order to prevent the spread of the epidemic from becoming invisible, and thus carry out the city's new coronavirus nucleic acid screening, which was conducted within the scope of the city's territory for all residents, and at the same time, closed public places were suspended, and residents were advised to reduce their outings. At the same time, closed public places suspended business, and advise residents to reduce outdoor activities, and the content of the circular similar to the above notice continued to be issued until the 162nd notice, during this period of Zhengzhou population Omicron infection rate is extremely low. On December 4, 2022, Zhengzhou City issued Circular No. 163, which made adjustments to the epidemic prevention and control measures, and resumed the opening of closed public places. On December 8, Circular No. 164 announced that it was no longer necessary for people to show their health codes and nucleic acid negative certificates when moving around, and that centralized quarantine would no longer be applied to positively infected people. Since then, the number of people moving around Zhengzhou has increased and social production has resumed. However, the proportion of people infected with omicron has increased dramatically, with a large number of uninfected residents becoming

infected for the first time, leading to a situation where the majority of people in Zhengzhou were actually still at home in December. This infection trend only slows down after January 2023 due to the herd immunization situation; after mid-January, when the herd immunization covers almost the entire population, the number of new infections decreases to a level that does not affect the normal production of society. Thus, our sampling period covers virtually the entire period of Omicron infection.

2. Line 145: Please provide references to other literature discussing the TH-PKU 300b instrument and its methodology.

2. Ambient VOCs were collected and plumed into refrigeration and pre-concentration system. Programmed increased temperature method was used to separate each VOC species. We have added this to the revised version of the manuscript, details of which can be found in Line 175-176

Line175-176: A detailed description of the instrumentation can be found in our previous study (Zhang et al., 2021; Shi et al., 2024; Zhang et al., 2024).

Reference:

Shi, Y., Liu, C., Zhang, B., Simayi, M., Xi, Z., Ren, J., and Xie, S.: Accurate identification of key VOCs sources contributing to $O_3$ formation along the Liaodong Bay based on emission inventories and ambient observations, Science of the Total Environment, 844, 156998, 10.1016/j.scitotenv.2022.156998, 2022.

Zhang, D., He, B., Yuan, M., Yu, S., Yin, S., Zhang, R.: Characteristics, sources and health risks assessment of VOCs in Zhengzhou, China during haze pollution season, Journal of Environmental Sciences, 108. 44-57, 1001-0742,

https://doi.org/10.1016/j.jes.2021.01.035, 2021.

Zhang, D., Li, X., Yuan, M., Xu, Y., Xu, Q., Su, F., Wang, S., Zhang, R.: Characteristics and sources of nonmethane volatile organic compounds (NMVOCs) and $O_3$–$NO_x$–NMVOC relationships in Zhengzhou, China, Atmosphere Chemistry and Physics, 24, 8549-8567, https://doi.org/10.5194/acp-24-8549-2024, 2024.

3. Line 160: Was the instrument calibrated for all 160 compounds?

3. Thank you for your suggestion. After a comprehensive selection process, 106 VOCs were ultimately included in this study. To ensure the accuracy of the VOCs, all 106 substances were injected into the standard gas through a five-point standard curve. This methodology has been added to the manuscript. Should you have further suggestions, we kindly request that you contact us so that we can carefully revise the manuscript according to your suggestions.

4. Line 252: Jan 1 as an absolute cutoff of "infection period" and "recovery period" needs proper debate.

4. Since the lifting of the containment policy in early December 2022, there has been a notable increase in the number of Omicron infections in Zhengzhou City. The peak number of infections occurred in mid- to late-December, after which the rate of infections declined rapidly. Since January, the number of new infections has remained relatively stable. Clinical observations indicated that individuals without a history of other illnesses typically required approximately one week of rest following their initial Omicron infection. Additionally, the majority of residents had recovered

from their first infection and resumed their normal work activities by January. It seems reasonable to posit that January 1 marks the transition between the infection period and the recovery period.

Results and Discussion:

1. Figure 2: The composition pie chart does not show a discernible difference between the two cases. Surprisingly, the recovery period had even lower $NO_x$ and aromatics than the infection period. Please clarify.

Response:

1. Thank you for your suggestion. The decrease in the percentage of aromatic hydrocarbons is mainly caused by the increase in the percentage of OVOCs and halogenated hydrocarbons. From Table 2, we can see that the concentration of aromatic hydrocarbons is increased in the recovery period compared to the infection period, which is increased by 28%.

2. Line 286: What do you mean by "highest increase ratio"?

2. We apologize for any confusion caused by our presentation. It has been revised to read: Although aromatic hydrocarbons have the lowest volumetric contribution (6% in Case 1 and 7% in Case 2), they show the largest increase from clean days to pollution.

3. Tables 1 and 2: The criteria for dividing the infection period vs. the

recovery period are unclear. In several instances, the terms Cases 1 and 2 are used interchangeably with infection and recovery periods, which is confusing.

3. We apologize for the confusion in your reading. We have labeled the infection and recovery periods in the chart. The reason for the division can be found in the answer to the previous question. Here we have compared the pollution during the infection period with the recovery period, Case 1 and Case 2. This is to highlight the increase in air pollution after the end of the epidemic infection.

4. Line 303: The use of VOC ratios possesses uncertainty. If you reference values of 0.13-0.7 for combustion, how does an average value of 1 align with combustion? Similarly, how do you reconcile the isopentane/n-pentane ratio with liquid petrol if the average does not fall within the referenced range? The same question applies to the isobutane/n-butane ratio for natural gas. The current explanation suggests a limited understanding of VOC ratios.

4. We acknowledge that this method has limitations. This method is only a preliminary determination of the emission sources of VOCs, and this method has been used in several studies (Yu et al., 2021; Wang et al., 2023; Xu et al., 2023). The mean value of T/B in our case is 1, which is between combustion and transport emissions (1.3-3.0), and thus receives the

combined effect of combustion and transport emissions. This phenomenon is also found in previous studies (Wang et al., 2023).

Reference:

Wang, B., Liu, Z., Li, Z., Sun, Y., Wang, C., Zhu, C., Sun, L., Yang, N.: Characteristics, chemical transformation and source apportionment of volatile organic compounds (VOCs) during wintertime at a suburban site in a provincial capital city, east China, Atmospheric Environment, Volume 298, 119621, ISSN 1352-2310, https://doi.org/10.1016/j.atmosenv.2023.119621, 2023.

Xu, Z., Zou, Q., Jin, L., Shen, Y., Shen, J., Xu, B., Qu, F.: Characteristics and sources of ambient Volatile Organic Compounds (VOCs) at a regional background site, YRD region, China: Significant influence of solvent evaporation during hot months, Science of The Total Environment, Volume 857, Part 3, 159674, ISSN 0048-9697, https://doi.org/10.1016/j.scitotenv.2022.159674, 2023.

Yu, S., Su, F., Yin, S., Wang, S., Xu, R., He, B., Fan, X., Yuan, M., Zhang, R.: Characterization of ambient volatile organic compounds, source apportionment, and the ozone–NOx–VOC sensitivities in a heavily polluted megacity of central China: effect of sporting events and emission reductions, Atmosphere Chemistry and Physics, Volume 21, 15239-15257, ISSN 1680-7324, https://doi.org/10.5194/acp-21-15239-2021, 2021.

5. Figure 5: The figure caption is awkwardly written. Combine the note into the caption itself. Generally, wind speeds below 1 m/s indicate local emissions and are not suitable for CPF analysis. Apply a filter of <1 m/s for CPF. Despite CPF indicating local sources, you also mention long-range transport as the most influential source in another instance, which is contradictory.

5. Thank you for your suggestion. We have merged the notes into the title. We have applied a <1 m/s filter for CPF as per your suggestion and revised

the corresponding description in the text (Fig. 5). We extend our sincerest apologies if our choice of words has caused any confusion or misunderstanding. we have mentioned in other examples that long-range transport also contributes to emission sources, but is not identified as the most significant source, and we have revised this section.

[Figure]

Fig. 5. CPF plots of five VOCs sources obtained using the PMF model. Note: a: Fuel evaporation; b: Solvent usage; c: Industrial source; d: Vehicular emission; e: Combustion.

6. The study's main aim seems to be comparing infection vs. recovery cases and/or pollution Cases 1 and 2 vs clean period, yet the overall PMF for entire sampling is discussed in the main text, while PMFs differences for the periods of interest are in the supplementary section. This does not align with the study's objectives.

6. Thank you for your comments. A discussion of the PMF differences

between the infection and recovery periods and contamination Case 1 and 2 versus the clean period has been added to the manuscript. The PMF factor profiles for the relevant periods are shown in Figure S6 in the supplementary section.

7. Line 374: Clarify what is meant by "peak of Omicron infection period." The terminology used is confusing, making it difficult to follow the results.

7. We apologize for that our description in this section was not clear. We have changed the section to "Figure S6 compares the differences in PMF source profiles between the Omicron infection period and the recovery period, as well as between the pollution day and the clean day."

---

## Author Response (AR3)

Title: Simultaneous observations of peroxyacetyl nitrate and ozone in central China during static management of COVID-19: Regional transport and thermal decomposition.

Authors: Bowen Zhang[1,3], Dong Zhang[2,3], Zhe Dong[2,3], Xinshuai Song[1,3], Ruiqin Zhang[1,3], Xiao Li[1,3,*]

Manuscript number: egusphere-2024-575

**Dear Editor**

Thank you very much for your time and effort in revising this paper. We apologize for the issues that were not addressed in the revised manuscript. We checked each comment carefully and strive to provide a satisfactory answer. Changes made in response to these comments are marked in yellow in the highlighted copy of the revised version. Our own minor changes are highlighted in red.

The following is a point-by-point response to each reviewer's comments.

**Reviewer #1:**

your answer begins by implying that meteorology is most important since "emissions from pollutant sources usually change very little over a period of time" while meteorology does change, but later in the same paragraph you state that meteorological differences between the two periods are

"minor" (though the reader does not know what level of difference "minor" entails) and, without any evident justification, say that meteorology is not responsible for any differences between the pollution periods. It's also not clear what the sentence "when comparing the meteorological conditions of the two pollution processes, none of the processes showed a tendency to be more prone to pollution" means; what are the two "processes"? Overall, this section requires a more quantitative approach; by how much should temperature, wind speed, and humidity affect pollution levels (based either on theory or prior research in this area), and therefore, why can we be sure that the differences are not meteorological? It could be that the conclusion remains there's "no discernible trend towards greater susceptibility to pollution" (L280-281) in period 2, but that needs quantitative justification.

Response: Thank you very much for your suggestion. We have revised the presentation of the section:

Previous studies have shown that meteorological factors such as low WS, high RH, and low precipitation are responsible for the increase in $PM_{2.5}$ pollution in Zhengzhou in winter (Duan et al., 2019). Our analysis of the correlation between different pollutants and meteorological conditions during the pollution period showed that $PM_{2.5}$, TVOCs and $NO_x$ were positively correlated with relative humidity (Fig. S3), which is consistent with the results of some previous studies (Wang et al., 2019). Yu et al.

(2018) identified RH and WS as the most influential meteorological conditions of $PM_{2.5}$ during winter. Their findings revealed a positive correlation between hourly $PM_{2.5}$ concentrations and RH ($r = 0.84$, $p < 0.01$) and a negative correlation between $PM_{2.5}$ concentrations and WS ($r = -0.62$, $p < 0.01$). The WS and RH between the infection and recovery periods were similar in this study which were largely considered to be of the same type of weather (Yu et al., 2018). However, the mean $PM_{2.5}$ concentration during the recovery period was found to be 1.6 times higher than that observed during the infection period. Furthermore, the concentrations of other pollutants (including $SO_2$, $NO_2$, CO, and $O_3$) exhibited analogous trends during the infection and recovery periods. The concentration of TVOCs during the recovery period was 1.2 times higher than that during the infection period, exhibiting a significant upward trend following the resumption of production. It is notable that WS, which is only 0.3 m/s higher in Case 1 than in Case 2, and RH, which is 13% higher in Case 1 than in Case 2, were relatively stable, while the concentration of pollutants is significantly higher in Case 2 than in Case 1. This is presumably attributable to the resumption of production activities in Case 2, which resulted in a notable increase in emissions. Decreased trends of air pollutants were found in other studies before and after the outbreak of the novel coronavirus (COVID-19) in early 2020 (Qi et al., 2021; Wang et al., 2021).

**Reviewer #2:**

Line 110: This added text is extremely confusing and difficult to read. The tenses switch between present and past, and most of the sentences are run-ons with many connected clauses. It's not clear why details of the city's screening processes are needed for this manuscript. Ideally this paragraph could be clarified, substantially shortened, and boiled down to only the most salient aspects for the present analysis. It would also be helpful if statements about human behavior during the various periods (e.g. "the number of people moving around Zhengzhou has increased", "people basically rested at home", etc.) could be supported by citations to sources that corroborate this.

Response: I'm sorry for the problems that arose. We have simplified the paragraph and corrected the tenses. The revised content is as follows:

In this study, we conducted continuous online observations of VOCs during the polluted winter season at an urban site in Zhengzhou. The study covered the period following the removal of lockdown measures. We focused on pollution events when the daily average $PM_{2.5}$ concentration exceeded 75 μg/m³ (China's Class II standard) for more than three consecutive days. Days with $PM_{2.5}$ concentrations below 35 μg/m³ (China's Class I standard) were classified as clean days. During this period, China lifted zero-COVID strategies, announcing the '10 measures' for optimizing

COVID-19 rules on December 7, 2022 (http://www.news.cn/politics/2022-12/07/c_1129189285.htm, Accessed Jan 2024). Zhengzhou's epidemic prevention and control measures changed with the issuance of Circular No. 163 on December 4, 2022, which allowed the reopening of closed public places. As a result, movement within Zhengzhou increased and social production resumed. Our research specifically examines the period dominated by the COVID-19 Omicron variant. where they demonstrate notable differences from the early virus strains (i.e., original SARS-CoV-2 virus and Delta) in terms of geographical transmission, the scale of the infected population, and symptom manifestation (Petersen et al., 2022; Merino et al., 2023).

Results & Discussion question 1: your response does not address the reviewer's point that the $NO_x$ decreased in the recovery period.

Response: I apologize for omitting a response to this question due to my oversight. Thank you very much for raising it. The reason for the reduction of $NO_x$ in Zhengzhou during the recovery period (here the reviewer actually refers to Case 2) may be related to the reduction of motor vehicle travel. A few days before Case 2, the Zhengzhou Municipal People's Government initiated a severe pollution weather level II response (https://sthjj.zhengzhou.gov.cn/tzgg/7037130.jhtml) and introduced emission control measures for industrial and mobile sources. We counted

the concentrations of $NO_x$ at other monitoring sites in the urban area of Zhengzhou during Case 1 and Case 2 (Table 1). Compared with Case 1, both found that Case 2 $NO_x$ concentrations decreased somewhat.

Table 1. $NO_x$ value of other urban sites in Zhengzhou ($\mu g/m^3$)

| Monitoring station | Case 1 | Case 2 |
| --- | --- | --- |
| Station 1 | 111.0 ± 56.4 | 108.1 ± 52.2 |
| Station 2 | 117.0 ± 67.6 | 112.3 ± 70.5 |
| Station 3 | 113.0 ± 103.1 | 93.7 ± 55.8 |
| Station 4 | 115.1 ± 61.8 | 114.7 ± 66.9 |

on VOC ratios -- my interpretation is that the reviewer was not implying that the isomer ratio method has limitations, but that its application is incorrect here, and that hasn't been fixed in your revisions. For example, your measured ratio of isobutane to n-butane is squarely within the range for LPG, above the range for vehicle emissions, and below the range for natural gas emissions (L338-340). Possible interpretations of your measured value of 0.5 imply (a) a mix between all three sources, (b) a mix between vehicle and natural gas sources, or (c) emissions dominated by LPG. However, you conclude on L342 that your measured value implies a contribution from natural gas, without explaining how you got there, even

though that doesn't line up with the range in which the measured value falls. The analysis of isopentane/n-pentane ratios seems similarly problematic (L332-337); the measured ratio of 1.4 doesn't necessarily mean that pentane is "mainly" derived from liquid petrol and fuel evaporation, since the same ratio could be achieved by mixing sources from coal and vehicle exhaust in the proper ratio. The analysis in this section needs reworking.

Response: Thank you very much for your suggestions. We didn't understand the reviewer's comments before and have some issues with the use of VOC ratios. We have revised this aspect of the analysis and incorporated additional information into Fig. 3. We believe that this modification offers a more illustrative representation of the distribution of VOC ratios across a given source interval, thereby providing a more robust foundation for the conclusions presented in this section. The modifications are as follows:

The toluene-to-benzene ratio (T/B ratio) was widely used to assess the relative importance of different sources. Specifically, T/B ratio with a value of 1.3-3.0 was observed in vehicle emissions for vehicles with different fuel types (Schauer et al., 2002; Wang et al., 2015). The reported T/B ratio for combustion processes was between 0.13 and 0.7 (Li et al., 2011; Wang et al., 2014). The mean value of T/B ratio for the entire period was 1.0, with the majority of the data (99%) falling between 0.1 and 3.0 and

concentrated within the 0.7-1.3 range (49%). This suggests that both traffic emissions and combustion are significant sources of VOCs.

The isopentane/n-pentane concentration ratios of 0.6-0.8 represent mainly coal combustion emissions, ratios of 0.8-0.9 represent LPG emissions, 2.2-3.8 represent vehicle exhaust emissions, and 1.8-4.6 represent fuel evaporation (Conner et al., 1995; Liu et al., 2008; Li et al., 2019). The sources of isopentane and n-pentane in this study were intricate and multifaceted. The mean isopentane/n-pentane ratio was 1.4, with the majority of data points (99%) falling within the range of 0.1-4.6, with a notable concentration in the 0.8 to 1.8 interval. This indicates that pentane is influenced by a combination of emissions from LPG and fuel evaporation.

Isobutane/n-butane concentration ratios of 0.2-0.3 represent vehicle emissions, 0.4-0.6 represent LPG usage, and 0.6-1.0 represent natural gas emissions (Russo et al., 2010; Zheng et al., 2018). The mean isobutane/n-butane ratio in this study was 0.5, with the majority of data points (99%) falling within the 0.4-0.6 range, indicating that VOCs at the observation sites were significantly influenced by the use of LPG. (Shao et al., 2016; Zeng et al., 2023).

The ratio of X/E can be used to infer the photochemical age of the air mass. X/E ratios around 2.5-2.9 are typical of urban areas, indicating that VOCs

are mainly from the urban area (fresh air mass) (Kumar et al., 2018). When this ratio is significantly lower than 3.0, it indicates that VOCs are mainly transported from distant sources (aging air masses) (Kumar et al., 2018). The average X/E value in this study was 2.0 (Fig. 3(d)), indicating low photochemical activity and aging of the air mass at the observation site. Potential source analyses also indicate that air masses are affected by long-range transport (Fig. S4).

[Figure]

Fig. 3. Correlation analysis between specific VOC species.

---

## Author Response (AR4)

Title: Simultaneous observations of peroxyacetyl nitrate and ozone in central China during static management of COVID-19: Regional transport and thermal decomposition.

Authors: Bowen Zhang[1,3], Dong Zhang[2,3], Zhe Dong[2,3], Xinshuai Song[1,3], Ruiqin Zhang[1,3], Xiao Li[1,3,*]

Manuscript number: egusphere-2024-575

**Dear Editor**

Thank you very much for your time and effort in revising this paper. We apologize for the issues that were not addressed in the revised manuscript. We checked each comment carefully and strive to provide a satisfactory answer. Changes made in response to these comments are marked in yellow in the highlighted copy of the revised version. Our own minor changes are highlighted in red.

We are not sure if we fully understand your meaning. If there are any areas where the modifications are not reach the designated position, we hope you can provide further guidance. Your help was very much appreciated.

The following is a point-by-point response to each comment.

**Reviewer #1:**

First, while the section describing the influence of meteorology (3.1) is easier to read, the analysis was not made more quantitative in this revision as requested. I believe the fundamental concern here is that no effort has been made to calculate, quantitatively, how much the higher pollution in Case 2 could be due to meteorology rather than increased emissions. The current analysis brushes off any meteorological differences as minor, even though the higher wind speed and temperature and lower humidity might all contribute to a buildup of gas phase pollutants. For example, given that the wind speed in Case 2 is 25% lower than in Case 1, in a simple box model the pollutants could be expected to build up to levels $[1/(1-.25)-1]$ = 33% higher just due to the lower ventilation, which is enough to account for much of the concentration differences of some pollutants between the periods. The influences of temperature and humidity may be more complex, but still, a quantitative estimate of their impact would be useful, or at least an acknowledgment of their potential importance. In cases where this fundamentally changes the conclusions that can be drawn from the analysis -- for example, if the difference in wind speed is indeed sufficient to explain some appreciable portion of the difference in pollutant concentrations -- that should also be noted in the conclusions and abstract.

Response:

We would like to express our gratitude for your contributions to the discussion. Due to technical and methodological limitations, we are unable to quantitatively analyze the impact of meteorological factors on pollution at present, which will be studied in the future. However, we have reviewed the literature and expanded our description of the relationship between meteorology and pollutants, affirming the importance of meteorological factors in pollution formation according your suggestion.

Prior research has indicated that low WS and high RH are associated with elevated $PM_{2.5}$ concentrations in Zhengzhou during the winter season. As indicated in our manuscript, the RH in Case 2 is 12% lower than that in Case 1. This difference in meteorological conditions is indicative of a greater propensity for pollution in Case 1. However, the discrepancy in wind speed (0.3 m/s) between Case 1 and Case 2 is deemed to be of negligible magnitude during the observation periods, and the actual impact is inconsequential.

To illustrate the important effect of meteorological conditions on pollution generation, we included a comparison of the difference in meteorological parameters between clean and polluted days. In this study, the WS on clean days (1.4 ± 0.8 m/s) was higher than that in Case 1 (1.2 ± 0.9 m/s) and Case 2 (0.9 ± 0.7 m/s), while the RH was 26.2% and 12.5% lower compared to Case 1 and Case 2, respectively. Therefore, adverse

meteorological conditions play a significant role in the occurrence of pollution during the observation period.

At the same time, we have made appropriate modifications to the content and added descriptions and analyses in the manuscript to affirm the importance of meteorological conditions in pollution formation. (Lines 261-268)

Lines 261-268: In this study, the WS on clean days (1.4 ± 0.8 m/s) was higher than in Case 1 (1.2 ± 0.9 m/s) and Case 2 (0.9 ± 0.7 m/s), while the RH was lower by 26.2% and 12.5% compared to Case 1 and Case 2, respectively. These findings indicate that high RH and low WS significantly influence the occurrence of pollution during the observation period.

**Reviewer #2:**

Second, the interpretations about source apportionment in the VOC ratio analysis section (3.2) remain more conclusive than the data allow. I think the edits to Figure 3 have improved its usefulness. However, it is not true that the measured ratio, for example, of isopentane to n pentane "indicates that pentane is influenced by a combination of emissions from LPG and fuel evaporation". The same mean ratio could be reached through a linear combination of emissions from coal combustion and vehicle exhaust, for

example. The statements attributing emissions to particular sources in this section should be hedged accordingly, acknowledging that other interpretations of these data can't be ruled out.

Response:

Thank you for your comments. We realize that the conclusions here are not rigorous enough, and we must admit that alternative interpretations of these data cannot be ruled out. We have revised this part of the manuscript. (Lines 334-339)

Lines 334-339: The isopentane/n-pentane concentration ratios of 0.6-0.8 represent mainly coal combustion emissions, ratios of 0.8-0.9 represent LPG emissions, 2.2-3.8 represent vehicle exhaust emissions, and 1.8-4.6 represent fuel evaporation (Conner et al., 1995; Liu et al., 2008; Li et al., 2019). The sources of isopentane and n-pentane in this study were intricate and multifaceted. The mean isopentane/n-pentane ratio was 1.4, with the majority of data points (99%) falling within the range of 0.1-4.6, with a notable concentration in the 0.8 to 1.8 interval. This indicates that pentane is susceptible to a combination of LPG emissions and fuel evaporation. It should be noted that this analytical approach is not without limitations. For instance, the proportionality of pentane may be influenced by a combination of LPG emissions and fuel evaporation. Consequently, an in-depth examination of the sources of VOCs was conducted using the PMF

model in the next section.

---

## Author Response (AR5)

Title: The variations of VOCs based on the policy change of Omicron in traffic-hub city Zhengzhou

Authors: Bowen Zhang[1,3], Dong Zhang[2,3], Zhe Dong[2,3], Xinshuai Song[1,3], Ruiqin Zhang[1,3], Xiao Li[1,3,*]

Manuscript number: egusphere-2024-575

**Dear Editor**

Thank you very much for your time and effort in revising this paper. We apologize for the issues that were not addressed in the revised manuscript. The changes suggested in your current response were very clear, and we believe we have fully understood your suggestions and made the appropriate changes. Changes made in response to these comments are marked in yellow in the highlighted copy of the revised version.

The following is a point-by-point response to each comment.

On the influence of meteorology: the fundamental concern was that the impacts of meteorology on the analyzed pollution events have not been quantified, and therefore it cannot be ruled out that meteorology plays a major role (potentially as strong as or stronger than emissions changes) in the differences in pollution seen in the two analyzed events. The way this is currently addressed in the manuscript is by ignoring the meteorological

differences without giving valid reasoning: e.g. "However, the discrepancy in wind speed (0.3 m/s) between Case 1 and Case 2 is deemed to be of negligible magnitude during the observation periods, and the actual impact is inconsequential." How is 0.3 m/s determined to be "negligible" and its impacts "inconsequential"? As was highlighted in a previous round of reviewer and editor comments, this 0.3 m/s represents a 25% decrease, which could be responsible for a non-negligible 33% increase in pollution levels based on a simple box model framework. Required edits:

EITHER: quantify the potential effects of wind speed and other variables with, for example, a simple box model framework, and incorporate this into the analysis; in cases where this fundamentally changes the conclusions that can be drawn from the analysis -- for example, if the difference in wind speed is indeed sufficient to explain some appreciable portion of the difference in pollutant concentrations -- that should also be noted in the conclusions and abstract.

OR: remove unsupported statements such as the sentence quoted above about the magnitude of meteorological effects, and acknowledge -- here in the meteorology section, but also in the abstract and conclusions -- that meteorology may also play a role in differences between pollution events,

and the importance of that role was not determined here.

Response:

We are grateful for your perspicacious recommendations for amendments. After careful consideration, we have decided to remove the statements that were not supported by sufficient evidence. In addition, we have modified the meteorology section, as well as the abstract and conclusions, to recognize that meteorology may also play a role in differences between pollution events, but we have not identified its specific impact on differences between pollution events in this paper. We would like to quantify the potential effects of wind speed and other variables in further as your suggestions.  The revisions are described below:

Lines 256-277: Previous studies have shown that meteorological factors such as low WS, high RH, and low precipitation are responsible for the increase in $PM_{2.5}$ pollution in Zhengzhou in winter (Duan et al., 2019). Our analysis of the correlation between different pollutants and meteorological conditions during the pollution period showed that $PM_{2.5}$, TVOCs and $NO_x$ were positively correlated with RH (Fig. S3), which is consistent with the results of some previous studies (Wang et al., 2019). The comparisons of average concentrations of different periods between different periods are

presented in Tables 1 and 2. In this study, the WS on clean days (1.4 ± 0.8 m/s) was higher than in Case 1 (1.2 ± 0.9 m/s) and Case 2 (0.9 ± 0.7 m/s), while the RH was lower by 26.2% and 12.5% compared to Case 1 and Case 2, respectively. These findings indicate that high RH and low WS influencing the occurrence of pollution during the observation period, which should be further studied in further.WS, Temp and RH conditions during infection and recovery periods were generally similar, and meteorology may also have played a role in the differences between pollution events, but its specific influence was not determined here. The average concentration of $PM_{2.5}$ during the recovery period was 1.6 times the value during the infection period. Furthermore, the concentrations of other pollutants including $SO_2$, $NO_2$, CO, and $O_3$ all showed a similar trend between infection and recovery periods. The TVOC concentration during the recovery period was 1.2 times the value during the infection period, showing an obvious increase trend after resuming production. Decreased trends of air pollutants were found in other studies before and after the outbreak of the novel coronavirus (COVID-19) in early 2020 (Qi et al., 2021; Wang et al., 2021).

On the VOC ratio analysis: the added lines (334-339) do not clarify this analysis. Instead they further obfuscate it, since not *two* sentences in

close proximity both state that pentane is coming from a combination of LPG emissions and fuel evaporation, without acknowledging that coal combustion and vehicle exhaust may also contribute. Further, the X/E ratio looks suspiciously uniform -- was it really perfectly on the 2.01 : 1 line for almost all observations? Or are the observations clustered in the lower-left of graph 3d (near the origin) more scattered? If so, it would be more helpful to zoom in there.

Required edits:

EITHER: remove the paragraphs about the VOC ratios, as this analysis does not seem central to your conclusions (which are largely based on the PMF analysis instead).

OR: explicitly acknowledge in each of the paragraphs about a different pair of species that the observed ratio does not rule out linear combinations of other sources. In particular, in the paragraph about isopentane/n-pentane, it should be noted that the observed ratio does not rule out contribution from coal combustion and vehicle exhaust, and in the paragraph about isobutane/n-butane, it should be noted that the observed ratio, while consistent with LPG usage, could also be achieved by a combination of

vehicular and natural gas emissions.

Response: We regret any inconvenience caused by the remaining issues with the revisions. In accordance with your recommendation, we have explicitly acknowledged in each paragraph on different species that the observed ratios do not preclude the possibility of linear combinations from other sources. In addition, zooming in on the lower left part of Figure 3d still shows that almost all observations are very close to the 2.01:1 horizontal line. We have added the zoomed-in section to Figure 3d.

[Figure]

Fig. 3. Correlation analysis between specific VOC species.

---

## Author Response (AR6)

Dear Editor,

Thank you for the valuable feedback from you and the reviewers. Our paper has now entered the proofreading stage, and we have unanimously agreed to make the following adjustment to the author order.

We would like to designate the current second author, Dong Zhang, as a co-first author. Dong Zhang has played a crucial role in the initial conceptualization, the design of the analytical methodology, data processing, and quality control. Additionally, he has dedicated significant time and effort to improving the content and quality of the paper during multiple rounds of revisions. Therefore, with the agreement of all authors, we have decided to recognize Dong Zhang as a co-first author to reflect his significant contributions to this research.

Thank you again for your consideration and guidance.

Sincerely,

Bowen Zhang and co-authors